# NL-Eye: Abductive NLI for Images

**Mor Ventura**[1*]  **Michael Toker**[1]  **Nitay Calderon**[1]  **Zorik Gekhman**[1,2]

**Yonatan Bitton**[2]  **Roi Reichart**[1]

[1]Technion  [2]Google Research

## Abstract

Will a Visual Language Model (VLM)-based bot warn us about slipping if it detects a wet floor? Recent VLMs have demonstrated impressive capabilities, yet their ability to infer outcomes and causes remains underexplored. To address this, we introduce NL-Eye, a benchmark designed to assess VLMs' visual abductive reasoning skills. NL-Eye adapts the abductive Natural Language Inference (NLI) task to the visual domain, requiring models to evaluate the plausibility of hypothesis images based on a premise image and explain their decisions. NL-Eye consists of 350 carefully curated triplet examples (1,050 images) spanning diverse reasoning categories: physical, functional, logical, emotional, cultural, and social. The data curation process involved two steps—writing textual descriptions and generating images using text-to-image models, both requiring substantial human involvement to ensure high-quality and challenging scenes. Our experiments show that VLMs struggle significantly on NL-Eye, often performing at random baseline levels, while humans excel in both plausibility prediction and explanation quality. This demonstrates a deficiency in the abductive reasoning capabilities of modern VLMs. NL-Eye represents a crucial step toward developing VLMs capable of robust multimodal reasoning for real-world applications, including accident-prevention bots and generated video verification.[1]

## 1 Introduction

*Abductive reasoning* refers to the ability to infer and predict plausible outcomes or causes given a context scene Peirce et al. (1934); Fann (2012); Douven (2021). This reasoning skill is crucial for Visual Language Models (VLMs), as they are likely to become increasingly integrated into our daily lives (Yildirim et al., 2024; Anwar et al., 2024; Chiang et al., 2024; Shah et al., 2023). These models will be required to accurately monitor and interpret daily life scenes and correctly infer plausibility to prevent accidents and provide timely advice. For instance, would a bot warn us from slipping on a wet floor when there is no warning sign? or would it infer a missing pacifier as a cause of a crying baby?

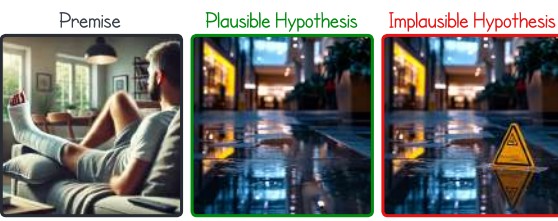

Figure 1: NL-Eye evaluates the abductive reasoning capabilities of VLMs. The main setup involves a premise image and two hypothesis images, where the model is tasked with inferring which hypothesis is more plausible, and to provide an explanation for its choice.

Although this capability is critical, previous work has mainly evaluated VLMs in a *single scene* setting — such as visual entailment or detecting improbable events like a fire in a closed jar — or in *sequential scenes*, such as next-frame prediction Xie et al. (2019); Fu et al. (2022); Hessel et al. (2022); Fu et al. (2024); Ganz et al. (2024); Yarom et al. (2024); Kadiyala et al. (2024). Consequently, it remains unclear to what extent existing VLMs are capable of abductive reasoning.

To address this, we introduce NL-Eye, a benchmark designed to evaluate *visual abductive reasoning* capabilities of VLMs across *multiple images*. NL-Eye is inspired by the textual abductive NLI

---

*Correspondence to: `mor.ventura@campus.technion.ac.il`

[1]Our data and code are available at `https://venturamor.github.io/NLEye/`.

task Bhagavatula et al. (2019) and applies it to the visual domain. In NL-Eye, a VLM is presented with a premise image and one or two hypothesis images. It then needs to infer how likely (plausible) a hypothesis image is to result from or lead to the premise image. The plausibility evaluation can be either done individually or in comparison to an alternative hypothesis. For instance, in Figure 1, the VLM needs to infer that, given the broken leg in the context image, it is more likely that the man slipped on the wet floor which lacked a warning sign (i.e., selecting hypothesis image 1).

Beyond *plausiblity prediction*, NL-Eye facilitates the evaluation of the models' capability to provide faithful explanations. This allows us to explore whether they are correct for the right reasons rather than relying on shallow heuristics McCoy et al. (2019). For example, a valid explanation for the broken leg scene would suggest that the presence of a warning sign would have made the man more alert, thereby potentially preventing the accident. In contrast, a shallow explanation might suggest that the man was simply resting on a cozy rainy day.

Each NL-Eye example features a premise image alongside two hypothesis images, annotated with a gold label indicating the index of the more plausible hypothesis. The examples also include a gold explanation detailing why the chosen hypothesis is more plausible than the alternative. Each example is categorized into one of six *reasoning categories* – physical, logical, emotional, functional, cultural, and social – and includes temporal annotations that specify whether the hypotheses occur *before*, *after*, or *simultaneously* with the premise, and whether the time duration between the premise and hypothesis scenes is *short* or *long*. This rich annotation aids in diagnosing current VLMs and highlights their strengths and weaknesses. Figure 2 presents a detailed example.

To create NL-Eye, we collected a large pool of high-quality textual scenes created by experienced human annotators. The resulting scenes were then provided to professional designers who utilized Midjourney and DALL-E (Ramesh et al., 2021) to synthesize the corresponding images. The designers are also tasked with categorizing each example and creating the explanation that is used as the gold label. The image generation process was iterative, requiring multiple attempts to ensure consistency between the textual descriptions and the visual scenes, as well as visual coherence among the images within the same triplet. This process resulted in a total of 1,050 generated images, yielding 350 image triplets. Overall, NL-Eye is characterized by carefully curated examples, offering high quality both in terms of the scenarios and the consistency and quality of the images.

The first analysis is *human evaluation* where annotators select the more plausible hypothesis and explain their choice. Our results indicate that humans successfully identify the more plausible hypothesis in $85\%$ of the cases. Furthermore, in our assessment of the quality of the human-generated explanations, we find that in $94\%$ of the cases where the correct hypothesis was selected, the humans also provided a valid explanation. This demonstrates that humans perform reasonably well on the NL-Eye tasks.

Next, we design a comprehensive study to evaluate the abductive reasoning abilities of modern VLMs. We take multiple measures to ensure the robustness of our evaluation, including addressing sensitivity to the order in which hypotheses are presented and exploring various input strategies, such as feeding the model three separate images or presenting it with a single combined-image that composites all three. Since real-world scenarios may not always provide two alternatives, we also evaluate the model's ability to assign a plausibility score to a single hypothesis, in addition to comparing two candidates. We have also developed a framework that utilizes a text-based baseline that processes textual descriptions of visual scenes. Specifically, we compare the results with gold descriptions and with the captions of the images as generated by the VLMs. Lastly, evaluating model-generated explanations is challenging, as comparing generated text to a single reference (gold) explanation can be limiting and may not capture the variety and validity of possible correct answers. To address this, we adopt the evaluation proposed by Bitton-Guetta et al. (2023): human annotators are presented with an image triplet where the correct hypothesis is already labeled and select valid explanations from a provided set.

Our results show that while humans perform well on NL-Eye, VLMs struggle, with most models failing to surpass a random baseline in the *plausibility prediction* task. Even when identifying the plausible hypothesis, VLMs fail to provide accurate explanations in over 50% of cases, revealing a major weakness in their abductive reasoning. Furthermore, our text-based experiments indicate that these models often succeed in textual reasoning even when they fail to reason over images. Interestingly, when we prompt the VLMs to generate image captions, the resulting captions prove

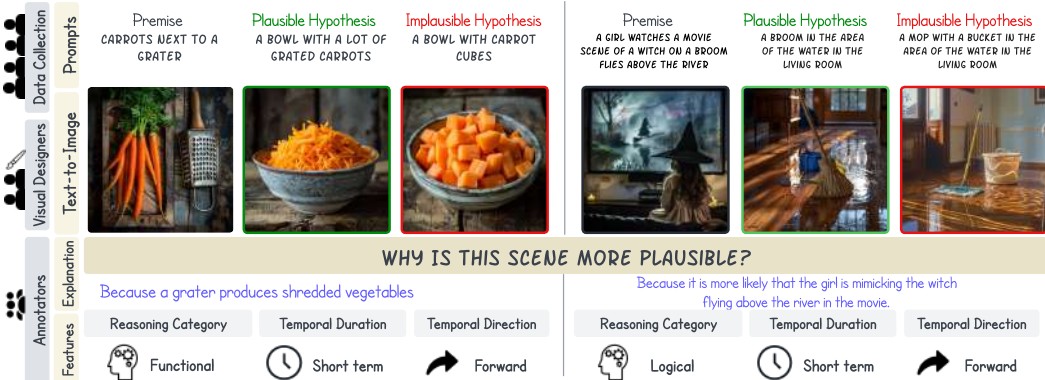

Figure 2: Fully annotated examples from NL-EYE. Each example includes the three images, the textual descriptions (prompts) used to generate them, the gold label, an explanation for why the gold is more plausible, and indications of the reasoning category and temporal direction and duration.

ineffective for solving the task. Consequently, we hypothesize that the VLMs reasoning is hindered by inaccurate visual interpretations. We also find that these models are sensitive to the order in which the hypotheses are presented and to the input format (three separate images vs a single combined-image). This sensitivity is concerning, as it raises the possibility that the models may not genuinely understand the underlying concepts, potentially relying on superficial cues to make decisions.

To summarize, we introduce NL-EYE a carefully curated benchmark designed to test the abductive reasoning abilities of VLMs across various categories and temporal relations. We then conduct a comprehensive study evaluating modern VLMs on NL-EYE and find notable deficiencies in their abductive reasoning capabilities. We believe NL-EYE represents a crucial step toward enhancing the reasoning abilities of VLMs, moving them closer to truly understanding complex, real-world scenarios and providing more reliable and interpretable outputs.

## 2 THE NL-EYE BENCHMARK

### 2.1 TASKS

Our objective is to explore and benchmark the abductive reasoning capabilities of modern VLMs. Unlike much of the previous work in NLP, our focus is on reasoning solely based on visual inputs: premise and hypothesis images. The *premise image* illustrates the context – factual observations about the world and a starting point from which conclusions are drawn. The *hypothesis image* illustrates a candidate conclusion – a possible event that could occur before, after, or simultaneously with the scenario presented by the premise image. In the context of our study, we refer to the definition of *abductive reasoning* Nie et al. (2020); Douven (2021) as a form of logical reasoning that seeks the most plausible hypothesis (conclusion) given a premise (a set of observations).

To perform visual abducting reasoning, the VLM should identify objects and their relationships within each image, understand the relationships between the images, and integrate this information to reason about the plausibility of the hypotheses (see §B.1 for detailed definitions of the sub-capabilities involved in performing visual abductive reasoning). We introduce two novel tasks to evaluate those capabilities: *Plausibility Prediction* and *Plausibility Explanation*. In the prediction task, the VLM is provided with the premise and hypothesis images. Its goal is to predict the plausibility of the hypothesis images or to determine which one is more plausible. We argue that VLMs should be capable of not only predicting plausibility but also providing a sensible explanation of their reasoning process. Therefore, in the explanation task, the model is also required to generate a free-form textual explanation justifying why the chosen hypothesis is plausible or at least more plausible than the other.

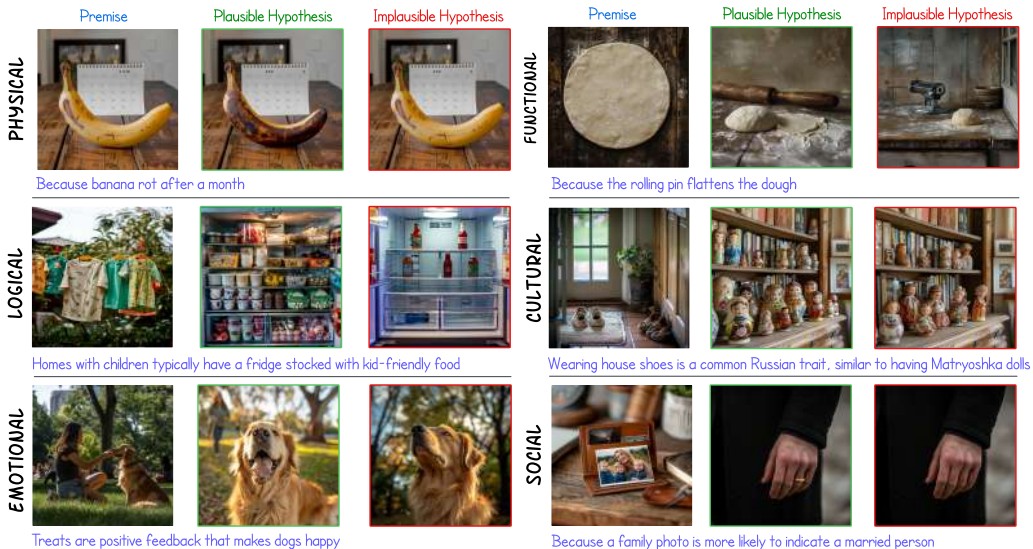

Figure 3: Real examples from each reasoning category in NL-EYE. The more plausible hypotheses are framed in green. The gold explanations are included below each sample.

## 2.2 BENCHMARK STRUCTURE AND CATEGORIZATION

In this subsection, we describe the structure of each example in our benchmark and discuss the taxonomy we proposed for categorizing the examples. In Figure 2, we present the structure of two examples, which contains: (1) the premise image; (2) two hypothesis images; (3) the label, which indicates the more plausible hypothesis and is given by the benchmark designers; (4) the textual descriptions of the three images that were used for generating the images; (5) the gold explanation, which clarifies why the correct hypothesis is more plausible, and is written by the benchmark designers; (6) reference explanations, which were written and validated by crowd-workers; (7) categorization of the example, which indicates the involved reasoning category, temporal direction and duration.

In §A, we describe the data creation process and specifically elaborate on components (1)-(5). The crowd-worker annotations (component 6) are detailed in §3.2. We next outline our proposed categorization, which serves two purposes: first, to ensure our benchmark is diverse, balanced, and covers a wide range of domains and reasoning types; and second, to aid in diagnosing areas where VLMs fall short.

**Reasoning Categories** We identify six different categories: *Physical*, *Logical*, *Emotional*, *Functional*, *Social*, and *Cultural*, ranging from physical reasoning (e.g., predicting the color of a rotten banana) to cultural reasoning (e.g., determining if a habit like wearing house shoes implies another cultural trait, such as owning Matryoshka dolls). Figure 3 presents an example from each category, with formal definitions in Appendix B.2.

**Temporal Categories** The temporal categories are based on direction and duration. Temporal direction refers to the logical relationship between the premise and hypothesis, indicating whether the event depicted in the premise image causes the hypothesis event (*forward*), is caused by it (*backward*), or if the events occur simultaneously (*parallel*). Examples that do not occur at the same time and are not categorized as parallel can also be classified by temporal duration, which is determined by the time gap between the events depicted in the premise and hypothesis images. These include *short-term* – when the events occur close in time, possibly in the same physical environment, or with no significant sequence of events separating them, and *long-term* – when the events take place in noticeably different periods of time. For instance, in the example on the left in Figure 2, the grated carrots suggest a short-term forward progression within the same environment as the whole carrot in the premise.

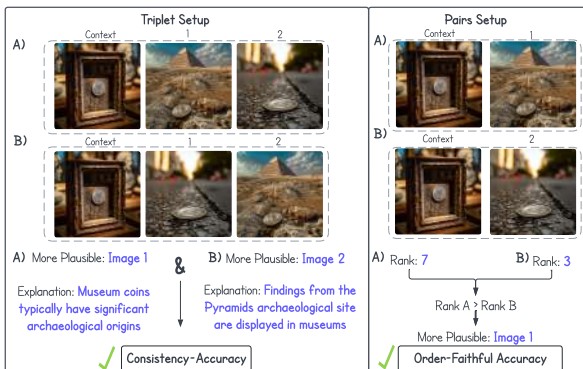

Table 1: Models and baselines by their input strategy and reasoning approach.

| Approach → Strategy → Model ↓ | Vision-Based | | Text-Based | |
|---|---|---|---|---|
| | Multiple Images | All In One | Image to Text | Text Only |
| Gemini-1.5-Pro | ✓ | ✓ | ✓ | ✓ |
| GPT-4 Vision | ✓ | ✓ | ✓ | ✓ |
| GPT-4o | ✓ | ✓ | | ✓ |
| Claude-3.5-Sonnet | ✓ | ✓ | ✓ | ✓ |
| Claude-3-Opus | | ✓ | ✓ | ✓ |
| Llava-1.6 | | ✓ | ✓ | ✓ |
| BLIP2-FlanT5-XL | | | ✓ | |
| InstructBLIP | | | ✓ | |
| BART-L-MNLI | | | | ✓ |
| DeBERTa-v3-nli | | | | ✓ |

Figure 4: **In the triplet setup** (left), the input of the VLM is a triplet of premise and two hypotheses images, and its task is to predict and explain which hypothesis is more plausible. We provide the triplet two times with different orders of the hypotheses (e.g., see A and B), and only if it is consistent and predicts the correct hypothesis for both we consider it an accurate prediction. **In the pairs setup** (right), the input is a premise and hypothesis, and the VLM should output a plausibility score. For the same premise and two hypotheses, the predictions of the VLM are considered order-faithful and accurate if the correct hypothesis is scored higher than the wrong one.

## 3 EXPERIMENTAL SETUP

### 3.1 TASKS AND SETUPS

Recall that the NL-EYE benchmark includes two tasks: *Plausibility Prediction* and *Plausibility Explanation*. Both tasks require reasoning about the relationship between the context image, the premise, which is denoted as $I_P$, and a candidate image, a hypothesis, denoted by $I_H$. In addition to the images, the model $f$ receives a textual query (a prompt) that contains instructions describing the task it should perform (see Appendix Tables 16 and 11). We introduce two setups for solving the tasks: the *Triplet* setup and the *Pairs* setup.

**The Triplet Setup** which is illustrated in the left box of Figure 4 the model receives the query along with three images: the premise ($I_P$) and two candidate hypotheses ($I_{H1}$ and $I_{H2}$). In the prediction task, the model's goal is to determine which of the two hypotheses is more plausible given $I_P$, i.e., which is more likely to occur, assuming $I_P$ is a true observation about the world. For the explanation task, the model is also required to generate a textual explanation justifying why the chosen hypothesis is more plausible than the other.

**The Pairs Setup** which is illustrated in the right box of Figure 4, the model $f$ receives the query and two images: the premise $I_P$ and the hypothesis $I_H$. The prediction task now is to provide a plausibility score that indicates how plausible $I_H$ is, given that $I_P$ is a factual observation. In our experiments, this score is provided on a 1-10 Likert scale. However, this is not mandatory – the plausibility score can be adapted to the needs of the model developer. For instance, the score could be expressed as a probability or another appropriate metric. In the explanation task, the model is asked to explain why $I_H$ can be plausible given $I_P$.

**Reasoning Approaches & Input Strategies** To thoroughly assess the abductive reasoning capabilities of current models, we use two reasoning approaches: *Vision-based* – where the model is tasked with performing the entire task end-to-end based solely on the visual input, and *Text-based* – where the final plausibility reasoning is based on textual input. In the vision-based approach, we experiment with two input strategies for feeding input to the model: (v.1) *combined-image* – where we concatenate the images (with the premise on the left) to form a single combined image, and (v.2) *separate-images* – where we feed in one prompt the images to the model separately, starting with the premise. As not all models support both strategies, Table 1 specifies which strategy was used for each model in our study. In the text-based approach, we utilize two input strategies as well: (t.1) *image-to-text* – where we ask the model to describe the two images in natural language, and then,

using those descriptions, the same model or another performs the plausibility prediction, and (t.2) *text-only* – we discard the visual inputs altogether and use only the textual descriptions generated when the images were created (see §A). The vision-based and text-based approaches allow us to understand the model's weaker capabilities better.

## 3.2 EVALUATION

**Predictions in the Triplet Setup**  At first, we evaluated models based on accuracy. However, we found that all models are sensitive to the positioning (in the all-in-one strategy) or the order (in the sequential strategy) of the hypothesis images, that is, whether $I_{H1}$ is placed or fed before or after $I_{H2}$. For example, models may perform differently when given Triplet A versus Triplet B from Figure 4. To address this sensitivity, we provide predictions for both orders of the hypotheses and then aggregate the two predictions. A prediction is considered correct if and only if the model selects the correct hypothesis in both orders. This approach reduces the likelihood of a correct prediction by chance and ensures the model demonstrates consistency. The performance score in the triplet setup is the described consistency accuracy (proportion of correct and consistent predictions; see §B.3).

**Predictions in the Pairs Setup**  In the pairs setup, we aim to evaluate the plausibility score predicted by the model. However, as previously mentioned, we do not want to constrain the model (or developers) to produce a specific score or adhere to a specific scoring function. Therefore, we do not support direct evaluation of the score, i.e., we do not provide a gold standard score against which the predicted score is compared. This raises the question: how do we plan to evaluate models in the pairs setup? The only assumption we require from the scoring function is *order-faithfulness* (Gat et al., 2024): if for a given premise $I_P$, the evaluated model $m$ scores one hypothesis $I_{H1}$ higher than another hypothesis $I_{H2}$, then $I_{H1}$ should genuinely be more plausible than $I_{H2}$. Accordingly, for every premise image, we take the two hypotheses and consider the scores of $f$ as correct if the hypothesis scored higher is the gold plausible hypothesis. The performance score in the pairs setup is the described order-faithfulness accuracy (proportion of correctly ordered scores; see §B.3).

**Human Evaluation of Explanations**  Evaluating free-text explanations is a challenging task due to the various ways explanations can be paraphrased and the reasoning involved in determining their validity. To address this, we followed the efficient human evaluation protocol proposed by Bitton-Guetta et al. (2023) and recruited crowd workers from Amazon Mechanical Turk (AMT). For each triplet of images, the workers were presented with the correct hypothesis and several explanations either written by humans (from the *human baseline* described in the next subsection) or generated by VLMs. We included only explanations of the correct hypothesis. Then, the workers were tasked to select all the explanations that are logic and justify why the correct hypothesis is more plausible (see Appendix E for additional details). We consider an explanation as correct if at least one worker selected it. The human evaluation score we present is the proportion of correct explanations.

**Automatic Evaluation of Explanations**  Through automatic evaluation, we aim to demonstrate a more scalable and cheaper approach to assessing the validity of model explanations. Like other recent works, we follow the common practice of employing an LLM as a judge (Zheng et al., 2023; Chen et al., 2024). Notice, that current models perform poorly on our visual abductive reasoning tasks, thus, expecting them to succeed in evaluating the validity of explanations generated by other models is pretentious. Instead, we simplify the task by conducting a reference-based evaluation, asking the LLM to determine whether the generated explanation aligns with a gold reference explanation – a task that relies solely on textual reasoning. The reference explanations include the gold explanations (see §A), augmented with human-written explanations approved during the human evaluation stage. To perform the automatic evaluation, we instruct an LLM (GPT-4o) to determine if the generated explanation aligns with one of the reference explanations (more details are provided in §I). The automatic evaluation score is the proportion of generated explanations that the judge LLM predicted as aligning with the references.

Table 2: **Main results:** Scores for vision-based experiments. Automatic evaluation scores are not presented for Humans since their explanations serve as references. Regardless of the input strategy, VLMs are greatly outperformed by humans and mostly perform on par or even below the baselines.

| Input Strategy | Model | Prediction | | Explanation | |
|---|---|---|---|---|---|
| | | Triplet | Pairs | Human | Auto |
| | **Humans** | 85% | 83% | 95% | — |
| **Separate Images** | Gemini-1.5-Pro | **51%** | 42% | 38% | 34% |
| | GPT-4-Vision | 46% | 40% | 39% | 37% |
| | GPT-4o | 16% | **50%** | 23% | 23% |
| | Claude-Sonnet-3.5 | 49% | 38% | **50%** | 26% |
| **Combined Image** | Gemini-1.5-Pro | 43% | 39% | 40% | 33% |
| | GPT-4-Vision | 41% | 34% | 37% | 27% |
| | GPT-4o | **60%** | **45%** | **44%** | 40% |
| | Claude-Sonnet-3.5 | 28% | 33% | 42% | 21% |
| | Claude-Opus-3 | 15% | 19% | 26% | 6% |
| | LLaVA 1.6 | 14% | 42% | 15% | 4% |
| | Fuyu | 4% | 44% | 10% | 2% |
| | **Random** | 25% | 45% | — | — |
| | **Dumb Pixel** | 50% | 50% | — | — |

### 3.3 MODELS AND BASELINES

Table 1 outlines the models used in our study, detailing their configuration, reasoning approach, and input strategy (specific versions in Appendix Table 15). Below, we provide more details on the models and baselines.

**VLMs**  We employ state-of-the-art closed source VLMs, including, Gemini-1.5-pro (Google, 2024), GPT-4-vision and GPT-4o (Achiam et al., 2023), and Claude-Sonnet-3.5 and Claude-Opus-3 (Anthropic, 2024). In addition, we employ open-source VLMs, including LLaVa 1.6 (Liu et al., 2024) and Fuyu (Bavishi et al., 2023).

**NLI models**  Recall that in the text-only reasoning approach, we provide the model with the gold text descriptions (used to generate the images) and ask it to predict which hypothesis description is more plausible. The predictor models we use in this approach include all the closed-source VLMs mentioned above, as well as fine-tuned NLI models such as DeBERTa-v3 (He et al., 2023) and BART-L (Lewis et al., 2019). When the predictor is an LLM, we ask it to determine which hypothesis description is more plausible given the textual premise description. For fine-tuned NLI models, we compute two 'entailment' scores between the premise and each hypothesis, and the final prediction is made based on the hypothesis with the higher score.

**Random baselines**  We present two simple baselines. The first is the *random baseline*, which randomly selects a hypothesis in the triplet setup or assigns a random score in the pairs setup. However, it is inconsistent due to its sensitivity to hypothesis order. To improve consistency, we introduce the *dumb pixel baseline*, which selects a hypothesis or assigns a score based on a predefined rule using the upper-leftmost pixel. For example, the hypothesis with the brighter pixel is deemed more plausible, or the score is calculated from the pixel's value.

**Humans**  Currently, there are indications that models can perform inference tasks at a level comparable to, or even exceeding, that of humans. Accordingly, we want to investigate whether these VLMs can match human performance on our tasks that appear straightforward for humans and expect them to succeed. To this end, we recruited 15 crowd-workers on the AMT platform. Pre-qualifications for workers were high approval rates and English-speaking countries. Additional details and guidelines are provided in §E.

# 4 RESULTS

## 4.1 VLMS FAIL TO PERFORM ABDUCTIVE REASONING WITH IMAGES

Table 2 presents the performance of humans, VLMs, and baselines on both tasks, prediction and explanation, for different setups and input strategies. For detailed results, refer to Appendix §C.1 and §C.2, where we provide extended experiments, as well as §D, where we compare human performance to VLM performance and analyze their alignment.

**VLMs Fail Where Humans Excel**    The results reveal a large performance gap between humans and VLMs on both tasks. Except GPT-4o, which achieves 60% accuracy in the triplet setup for combined-image inputs, *all VLMs perform worse than the dumb pixel baseline*. The situation is even more concerning for *current open-source VLMs, such as LLaVA 1.6 and Fuyu, which score below random baselines* (see additional open-source VLMs in Appendix Table 9). In contrast, human participants achieve 83-85% accuracy in the prediction task and 95% in the explanation task. Notably, the participants are crowd-workers who are not experts or highly skilled. This suggests that the task is neither unsolvable nor particularly difficult. Rather, current VLMs lack the visual abductive reasoning capabilities necessary to solve it effectively. Importantly, the finding that *tasks easily handled by humans pose significant challenges for VLMs* underscores the relevance of our benchmark and highlights areas where the research community can focus its efforts. Moreover, we found that VLMs are better in comparative or relative judgment setups (triplet, selecting which hypothesis is more plausible than the other) than in absolute judgment (pairs, predicting a plausibility score for a single hypothesis). This is unsurprising, as it is a known and well-studied phenomenon of humans (Pollitt, 2012; Verhavert et al., 2019) which was also observed in LLM-as-a-judge tasks (Kim et al., 2024). To the best of our knowledge, we are the first to show it for visual reasoning tasks.

**Even When Correct, VLM Explanations Are Unhelpful**    To assess the quality of the explanations, we conduct human and automatic evaluations. Since nearly half of the predictions are incorrect, we focused only on explanations for correct predictions, ensuring we can determine whether an explanation is genuinely poor rather than simply a result of the model failing to predict the correct answer. Note that in §D we provide a qualitative analysis of explanations of wrong predictions to better understand why they fail. Table 8 in the Appendix reports the number of evaluated explanations of each model (a total of more than 3,800), and the results are reported on the two rightmost columns of Table 2 (see human votes distribution in Appendix Table 7). As can be seen, humans almost always produce correct explanations, as 95% of the explanations were selected by the annotators. On the other hand, at best, only half of the explanations are selected. This demonstrates that *even when the VLMs predict correctly, the explanation is unhelpful*. In addition, we used valid human explanations as references for the automatic evaluation, which produces scores similar to those of the human evaluation in most cases. The automatic evaluation suggests that *VLMs produce explanations that describe different reasoning than humans*.

## 4.2 REASONS FOR FAILING TO REASON

This subsection presents experiments analyzing why VLMs fail at visual abductive reasoning.

**VLMs Can Perform Textual Reasoning – The Failure is in Visual Interpretation**    Table 3 presents the results of text-based experiments aimed at decoupling the textual reasoning capabilities from the visual ones. In the text-only approach, the models are provided with gold-standard descriptions of the images. As shown in the table, the performance of all models, including smaller fine-tuned NLI models, is significantly higher than their performance in vision-based experiments. Strong VLMs, such as GPTs and Claudes, achieve around 80% accuracy, nearing human performance. This indicates that *VLMs are capable of textual reasoning*. This finding suggests that the reasoning challenge does not mainly lie in the VLM's textual components but in the visual ones. In open-source VLM architectures (Liu et al., 2024), inference is not performed directly over images. Instead, the models encode images to latent visual representations, which are then passed to the language model component. We hypothesize that poor visual inference results from these visual representations being inaccurate for the reasoning task.

Table 3: **Text-Based:** Performance for prediction in the triplet setup. Predictor models perform well and similarly to (vision-based reasoning of) humans when using the gold description. However, VLM describers generate useless captions which do not help solve the task.

| Reasoning Approach | Describer | Predictor | Prediction Triplet |
|---|---|---|---|
| **Text-Only** | Gold | Gemini-1.5-Pro | 66% |
| | | GPT-4o | **80%** |
| | | GPT-4 | **78%** |
| | | Claude-Sonnet-3.5 | **79%** |
| | | Claude-Opus-3 | **81%** |
| | | BART L mnli | 68% |
| | | DeBERTa nli v3 | 65% |
| **Image-to-Text** | Gemini-1.5-Pro | GPT-4o | 29% |
| | GPT-4 vision | | 32% |
| | Claude 3.5 | | **39%** |
| | Claude 3 | | 33% |
| | LLaVA 1.6 | | 29% |
| | BLIP 2 | | **40%** |
| | Instruct BLIP | | 35% |

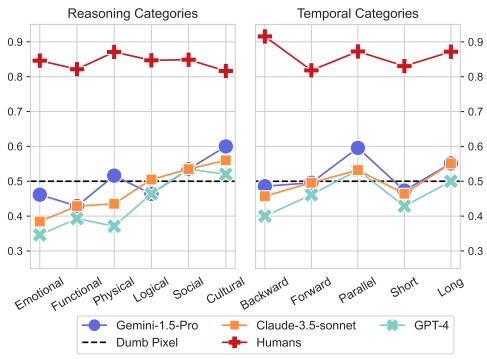

Figure 5: Vision-based performance with separate images for different reasoning categories (left) and temporal categories (right). VLMs struggle with the Emotional and Functional categories but perform better on Social and Cultural ones and on parallel reasoning.

In contrast, when VLMs generated descriptions for each image, and these descriptions were used as input to GPT-4o, instead of the gold ones, the performance dropped significantly, aligning with the results from the vision-based experiments (see Appendix Table 10 for Claude 3.5 as the predictor). We hypothesize that this indicates a *recognition gap*, where the generated descriptions either lack sufficient detail to capture the necessary information or are overly detailed (see examples in Appendix Figure 7), making it challenging to reason effectively.

Finally, consider the performance gap in Table 2 between separate and combined image input strategies. Except for GPT-4o, which consistently predicts the first hypothesis as more plausible (see Table 5 in the Appendix), the other three models perform better when using separate image inputs, showing an average improvement of over 10% (48.6% vs. 37.3%). This suggests that when an image is complex and contains many details, as in the case of a combined image, *VLMs struggle to encode the necessary details and represent each image successfully*.

**VLMs Predictions Depend on Hypothesis Location** Another factor contributing to the low performance of VLMs in the triplet setup is their lack of consistency. As shown in Table 5 in the Appendix, all VLMs are highly sensitive to hypothesis order, with performance variations ranging from 5-80%. For example, Gemini-1.5-pro's performance in the combined-image strategy drops from 82% when the second hypothesis is correct to 46% when the first hypothesis is correct. Similarly, GPT-4o, which performs best in the combined-image strategy (60%), fails in the separate-images approach, scoring 97% when the first hypothesis is correct but only 16% for the second. This suggests GPT-4o almost always predicts the first hypothesis as correct. In contrast, VLMs show much greater consistency in text-only inputs, with performance variation limited to 7%. *This indicates that VLMs rely on weak visual encoding, capturing superficial patterns like hypothesis order rather than meaningful image content*.

**VLMs are Better in Correlational and Knowledge-based Reasoning Compared to Causal Reasoning** Relying on the categorization within our benchmark, we present in Figure 5 the vision-based results in the triplet setup for six reasoning categories (left plot) and five temporal categories (right plot). VLMs exhibit a clear dichotomy in their reasoning abilities, excelling in some areas while falling short in others. Interestingly, the patterns are consistent across models, yet they diverge from the performance patterns observed in humans. For example, *VLMs perform best in Social and Cultural reasoning, where specific knowledge is key to correctly solving those examples*, see Ventura et al. (2023) for extended discussion about VLMs and cultural knowledge. In contrast, humans perform worst in the Cultural category. Another interesting observation is that VLM performance on parallel reasoning examples is higher than on forward and backward reasoning tasks. Notably, parallel reasoning may require only understanding correlations between the premise and the hypothesis, whereas forward and backward reasoning require causal reasoning – identifying causes (in

backward) or effects (in forward). This suggests that *VLMs may be more adept at identifying correlations rather than causal understanding*. Finally, the weakest category for VLMs is Emotional, which aligns with the literature (Lissak et al., 2024).

## 5 RELATED WORK

Recent advances in multimodal learning have enabled models to integrate textual and visual data across diverse tasks (Voulodimos et al., 2018; Guo et al., 2022; Qin et al., 2024). Powerful visual encoders like CLIP (Radford et al., 2021; Cherti et al., 2023) and SigLip (Zhai et al., 2023), coupled with the progress in LLMs (Chowdhery et al., 2023), have given rise to sophisticated VLMs such as BLIP2 (Li et al., 2023b), GPT-4 (Achiam et al., 2023), and Gemini (Google, 2024). These VLMs are pushing the boundaries of multimodal capabilities, tackling tasks like visual question answering (VQA) (Antol et al., 2015) and visual entailment (VE) (Xie et al., 2019). Our work builds on and extends research in these areas, with an emphasis on commonsense reasoning.

**From Textual to Visual Entailment** A cornerstone of our work is the expansion of Natural Language Inference (NLI), traditionally a text-based task (MacCartney, 2009; Dagan et al., 2010; Gekhman et al., 2023), into the visual domain. While previous research has explored NLI in the context of image-text alignment (e.g., SNLI-VE (Xie et al., 2019), TIFA (Hu et al., 2023), WYSI-WYR (Yarom et al., 2024), Mismatch-Quest (Gordon et al., 2023)), and even video-text entailment (Xu et al., 2021; Bansal et al., 2024), we introduce a novel framework for **image-to-image entailment**. This framework goes beyond simply selecting plausible alternatives by requiring models to explain their choices, thus offering a deeper evaluation of their abductive reasoning. Furthermore, we introduce a "pairs" setup that requires scoring the plausibility of image pairs, aligning our task more closely with the original formulation of textual entailment.

**Synthetic Data for Multi-Image Reasoning** Our work uniquely employs **synthetic images** generated by models like DALL·E (Ramesh et al., 2022), allowing greater control over visual complexity and diversity compared to natural image datasets such as Winoground (Thrush et al., 2022), Sherlock (Hessel et al., 2022), and VCOPA (Yeo et al., 2018). By emphasizing **multi-image reasoning**, we address limitations in existing datasets that focus primarily on single-image tasks, like WHOOPS!(Bitton-Guetta et al., 2023) and Visual Riddles(Bitton-Guetta et al., 2024). Our approach complements research on synthetic image understanding (Gokhale et al., 2022; Wu et al., 2023; Stöckl, 2023) and is better suited for commonsense reasoning in real-world contexts, enhancing datasets like SEED-Bench (Li et al., 2023a) and MMToM-QA (Jin et al., 2024), which tackle different aspects of multi-image reasoning. Unlike ScienceQA (Lu et al., 2022) and NTSEBench (Pandya et al., 2024), which focus on diagrams and scientific domains, our dataset employs photorealistic scenes from everyday life, making it more appropriate for evaluating commonsense reasoning. This integration of synthetic data, multi-image reasoning, and image-to-image entailment establishes a new benchmark for assessing VLMs' reasoning capabilities.

## 6 CONCLUSION

We introduced NL-EYE, a benchmark designed to assess the visual abductive reasoning capabilities of VLMs across multiple images. This skill is essential for real-world applications, such as accident-prevention bots. We paid special attention to detail in order to ensure that NL-EYE consists of high-quality and challenging examples, which required extensive human involvement at every stage of its curation. Our carefully designed study highlights critical challenges faced by modern VLMs in delivering satisfying plausibility predictions. We demonstrate that although humans perform well on these tasks, VLMs struggle significantly. This indicates a significant limitation of current models' ability to integrate visual interpretation with logical reasoning. Furthermore, models not only struggle to make correct predictions but also often fail to consistently provide helpful explanations. In future work, we would like to address these gaps, building on our insights to develop new VLM architectures with higher reasoning skills, mirroring human cognitive processes in complex environments, as elaborated in §F.

## ETHICS STATEMENT

The NL-EYE benchmark includes AI-generated images, with the potential presence of unpleasant or insensitive content. While we strive to minimize harmful biases, the inclusion of reasoning based on common sense knowledge and cultural perspectives may introduce further bias, particularly related to social norms. Additionally, the labels in this benchmark are based on consensus from human annotators, whose judgments may be influenced by their own cultural backgrounds, which could amplify bias. We also recognize the challenges related to text-to-image (TTI) copyrights, where the ownership of AI-generated content remains unclear. Researchers are encouraged to carefully consider these ethical and legal concerns when utilizing the benchmark.

## REPRODUCIBILITY

To ensure the reproducibility of our results and promote further research, we will publicly release the NL-EYE benchmark, along with the code. Detailed technical instructions, as well as documentation on how to use and adapt the benchmark, will be provided in a publicly accessible repository. Additional technical details, including model versions and specific configurations used in the experiments, are available in the Appendix (§I). By sharing these resources, we aim to foster transparency and support the research community in advancing the evaluation of VLMs.

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

# A  DATA CURATION

Joining recent efforts in evaluating VLMs with an emphasis on the quality of test sets over their sheer size (Thrush et al., 2022; Bitton-Guetta et al., 2024; Bitton et al., 2023; Bitton-Guetta et al., 2023; Padlewski et al., 2024), we carefully curated 350 test set examples. The creation process of NL-EYE required human involvement at every key step (see Figure 6), enabling the creation of diverse, high-quality examples tailored to the evaluation's specific goal.

NL-EYE is a multi-image benchmark consisting of daily life scenes. A "scene" refers to a specific setting where objects, people, and actions are arranged in a particular context, which can be represented either textually or visually. The benchmark includes both representations, and the following key steps in its creation process: (1) textual description writing, (2) image generation, and (3) explanation and categorization.

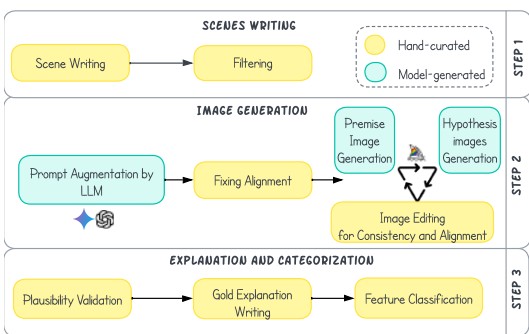

Figure 6: NL-EYE data curation workflow scheme. The process includes three steps: (1) writing textual descriptions, (2) generating images, and (3) generating explanation and categorization. Yellow denotes steps that require human involvement while turquoise denotes model-based generations.

**Textual Descriptions**   Scenes were manually crafted by a group of 20 annotators who were tasked with creating triplets consisting of a premise scene, and two hypothesis scenes, while one is more plausible than the other (see the first step, "scenes writing", in Figure 6). Each annotator had the flexibility to develop hypotheses across diverse reasoning categories, time directions, time durations, and domains. Annotators' creativity and experiences generated unique, everyday scenes that are often undocumented or scattered, making it hard to gather automatically similar data. We manually filtered scenes from the suggested pool based on several key criteria: (1) *premise necessity*, ensuring the scene is essential for determining the more plausible hypothesis; (2) *visual relevance*, guaranteeing the scenes can be effectively communicated visually; and (3) *uniqueness*, verifying that we do not replicate similar existing examples or logical patterns (see examples in Appendix Table 14). We also applied preferences for receiving a range of challenges, from easy to difficult, as well as diverse time shifts, including varying directions and durations. After applying these filters, we retained 75% of 450 suggested ideas.

**Image Generation**   The images in NL-EYE were manually curated by two of the authors (noted as "the designers"), who have experience with text-to-image models. This careful generation process ensures high-quality images and verifies consistency, alignment between text and images, and overall clarity. The images were generated based on the textual descriptions using mainly Midjourney and DALL-E 3 (Ramesh et al., 2021). During the *prompt augmentation* phase, The designers had the option to utilize assistance from Gemini and GPT-4 (Achiam et al., 2023) to transform the human-generated concise descriptions from step 1, into more detailed prompts with specific visual elements, enhancing visual consistency (see Appendix §H). For example, the step 1 textual description, *the boy is crying*, turns to the augmented prompt, *the curly redhead boy with the striped green t-shirt is crying*. Once the revised prompts were verified to ensure they don't interfere with the essential content (e.g., a change such as *the teenager is crying* or an omission such as *the curly redhead boy is wearing striped t-shirt*) and manual edits were made if necessary, the prompts were ready for image generation.

Typically, the process begins with generating the premise image. Image generation is an iterative process, involving repeated cycles of manual editing and image-to-image alignment until high quality and consistency are achieved. The image generation phase produces photorealistic images that are visually consistent, meaning that objects, people, and environments appearing in one image of the triplet are the same as in the others. The last guideline arises from the crucial need to exclude style from reasoning considerations in the future evaluation of VLMs on the task. Technically, visual consistency is achieved not only through prompt augmentations but also via inpainting (i.e., edit-

ing a specific region of an image using a textual prompt), image-conditioned prompting (generating an image while conditioning on another image), and using the same seed (initial noise distribution number) for all triplet images. See an example of image generation, step 2, in Figure 17 in Appendix §H.

**Explanation** For each example, the designers wrote gold-standard explanations [2] The gold explanation represents the original reasoning behind the scenes at the time of their curation. The gold explanation clarifies why the correct hypothesis is a more plausible outcome or cause of the premise. It often follows a pattern like *"Usually, X tends to Y"* or *"Because X made Y to..."*. Naturally, the explanation written at this stage is not the only possible explanation of the reasoning. Humans can suggest multiple plausible explanations and stories to justify connections between observations, also even for less likely scenarios. For example, examining the example of the *Social* reasoning category in Figure 3 (bottom-right row), the premise image depicts *a wallet with a family photo*, the hypothesis images depict *a man's hand with a wedding ring* (plausible) and *a man's hand without a wedding ring* (less plausible), and the human explanation is *a man with a family photo in his wallet is socially and statistically more likely to be married rather than single*. However, the man might be married but not wearing a ring, have a family without being married, or the scenes might tell a story of loss and remembrance.

**Validation and Categorization** The validation process consists of two key checks: (1) image-text alignment and (2) plausibility validation. First, we ensured that the images were correctly aligned with their corresponding texts. Second, we qualitatively assessed each example's plausibility, evaluating how difficult it is to understand the intended meaning and make any necessary adjustments. This process includes manual verification by the designers, supported by a *human baseline* (§3.3) with a high human success rate on the *plausibility prediction* task (85%; see §4) and strong inter-annotator-agreement (67.6%; unanimous votes) confirms the clarity.In addition, the designer classified the examples into relevant categories, as outlined in the previous section.

**Dataset Statistics** NL-EYE is categorized by reasoning categories, domains and temporal information. The *Logical*, *Social*, *Physical*, *Cultural*, *Functional*, and *Emotional* reasoning categories comprise 28%, 24.6%, 17.8%, 14.3%, 7% and 8%, respectively, of the benchmark examples. 78% of the examples are in the time duration of *short-term*, divided mainly (68%) with the *forward* direction. The *long-term* examples are 22% while 27% of them are also associated with the *backward* direction. Refer to the histogram of reasoning categories and real NL-EYE examples in Appendix Figures 18 and 16, respectively.

## B  EXTENDED DEFINITIONS

### B.1  SUB-CAPABILITIES OF VISUAL ABDUCTIVE REASONING IN NL-EYE

We propose that visual abductive reasoning can be organized into three levels of sub-capabilities, ranging from basic to advanced skills. Below, we outline and define these levels:

**Basic Capabilities**

- **Visual Recognition:** Comprehending visual elements and their spatial relationships within a scene.
- **Object Detection:** Identifying and recognizing objects in the visual input.

**Intermediate Capabilities**

- **Handling Multiple Images or Details:** Analyzing information across multiple images or managing intricate details within a single image.
- **Object Tracking:** Identifying an object in one image as the same object in another, even across changes in perspective or context.

---

[2]The benchmark contains the gold explanations and additional explanations written and validated by annotators, see §3.3.

**Advanced Capabilities**

- **Temporal Understanding:** The ability to interpret the sequence, direction, and duration of events in visual inputs. This includes identifying the flow of actions or changes, understanding their temporal order, and estimating how long they persist.

- **Order Sensitivity and Correlation Handling:** Recognizing the importance of sequence and distinguishing causal relationships from spurious correlations.

- **Common Sense Reasoning:** Applying general world knowledge to infer logical conclusions about the visual input.

- **Plausibility Assessment and Decision-Making:** Evaluating the likelihood of a scenario based on visual and contextual clues, and selecting the most plausible explanation or outcome from multiple options.

This hierarchical organization clarifies the definition of abductive reasoning in the context of NL-EYE.

## B.2  REASONING CATEGORIES DEFINITIONS

Table 4: NL-EYE textual descriptions examples. One example for each reasoning category. Every example consists of a premise phrase, a plausible hypothesis phrase, and an implausible hypothesis phrase.

| Category | Premise | Plausible Hypothesis | Implausible Hypothesis |
|---|---|---|---|
| Physical | *A child sits on the floor, holding a wrapped present in the shape of a rectangular box.* | *A child sits on the floor, holding an unwrapped rectangular present.* | *A child sits on the floor, holding an unwrapped ball-shaped present.* |
| Logical | *Clothesline with large shirts and small children's shirts.* | *A refrigerator full of home-made food, yogurts, and children's food.* | *An empty refrigerator with only a few bottles of beer and ketchup.* |
| Emotional | *A baby stroller with a paci-fier lying on the floor next to it.* | *A crying baby sits in a stroller.* | *A happy baby sits in a stroller.* |
| Functional | *A large thin circle of dough on a kitchen surface.* | *A lump of dough and a rolling pin on a kitchen surface.* | *A lump of dough and a pasta maker on a kitchen surface.* |
| Cultural | *A digital clock shows 16:00 pm and an image of Queen Elizabeth is on the wall.* | *British old ladies sit and drink hot tea cups.* | *British old ladies play contract bridge game.* |
| Social | *A person with a kiss mark on the cheek sitting at a holiday table with family.* | *Grandmother arrived as a guest.* | *Grandfather arrived as a guest.* |

**Physical Reasoning.**   Involves understanding the physical world and how objects interact within it. The scenes include changes in temperature, phase, shadow's location, color, shape, etc. This reasoning is inspired by the spatial-temporal reasoning definition (Deza et al., 2009).

**Functional Reasoning.**   Requires an understanding of objects' functionalities and tools' common usage. This type of reasoning involves not just recognizing objects and tools, but also comprehending their intended purposes and how they interact within various contexts. For instance, a hammer is not merely identified by its shape but also by its function of driving nails into surfaces. Functional reasoning allows a model to infer the appropriate use of an object within a given scenario, such as using a knife for cutting or a broom for sweeping.

**Social Reasoning.** Understanding social norms, relationships, and interactions. Social reasoning allows for the comprehension of social norms and etiquette, such as knowing how to greet someone depends on the context. This includes recognizing familial roles, friendships, professional relationships, and the varying degrees of formality and familiarity in interactions.

**Emotional Reasoning.** Understanding and interpreting emotions and emotional responses. It refers to the ability to identify a wide range of emotions, including happiness, sadness, anger, fear, surprise, and disgust, and to understand the context in which these emotions arise.

**Cultural Reasoning.** Involves acknowledging cultural traits and traditions while correctly associating them with their respective cultures. It includes the ability to recognize and interpret cultural symbols, rituals, languages, and behaviors accurately. For instance, it includes understanding that certain gestures may have different meanings in different cultures or that specific holidays and celebrations are unique to particular cultural or religious groups.

**Logical Reasoning.** Requires an understanding of general processes and broad commonsense. It enables the analysis of situations, draw inferences, and make decisions based on logical principles and widely accepted knowledge. Logical reasoning involves the ability to follow a sequence of steps to solve problems, recognize patterns, and identify relationships between different pieces of information.

### B.3 ACCURACY METRICS: MATHEMATICAL FORMULATION

Here we present the mathematical formulation of the accuracy metrics, based on the notations in Section §3).

Formally, the *consistency-accuracy* (*triplet accuracy*) is:

$$\text{consistency Acc.}(I_P, I_{H1}, I_{H2}, H_{gold}) =$$
$$\begin{cases} 1, & \text{if} \quad f(I_P, I_{H1}, I_{H2}) = f(I_P, I_{H2}, I_{H1}) = H_{gold} \\ 0, & \text{otherwise} \end{cases}$$

Formally, the *order-faithful accuracy* (*pairs accuracy*) is:

$$\text{order-faithful Acc.}(I_P, I_{H1}, I_{H2}, H_{gold}) =$$
$$\begin{cases} 1, & \text{if} \quad f(I_P, I_{H1}) > f(I_P, I_{H2}) \quad \& \quad H1 = H_{gold} \\ 1, & \text{or} \quad f(I_P, I_{H1}) < f(I_P, I_{H2}) \quad \& \quad H2 = H_{gold} \\ 0, & \text{otherwise} \end{cases}$$

# C COMPLEMENTARY RESULTS

## C.1 RESULTS OF PLAUSIBILITY PREDICTION AND EXPLANATION

Table 5: *Plausibility prediction* results of the triplet setup. Order refers to the position of the correct hypothesis image in the input, whether it was presented first (order 1) or second (order 2).

| Baseline | Model | Consistency Acc. | Order 1 Acc. | Order 2 Acc. |
|---|---|---|---|---|
| Separate-Images | Gemini-1.5-pro | **50.57%** | 56.0% | 75.14% |
| | GPT-4-vision | 45.71% | 59.14% | 66.0% |
| | GPT-4o | 16.29% | 97.43% | 16.29% |
| | Claude-3.5-sonnet | 49.28% | 65.9% | 59.31% |
| Combined-Image | Claude-3-opus | 15.14% | 30.57% | 72.86% |
| | Claude-3.5-sonnet | 28.29% | 76.86% | 30.29% |
| | Llava-mistral-7b | 14.86% | 53.14% | 26.0% |
| | Gemini-1.5-pro | 42.57% | 46.29% | 81.43% |
| | GPT-4-vision | 41.14% | 54.57% | 67.14% |
| | GPT-4o | **60.0%** | 76.0% | 69.14% |
| | Fuyu-8b | 4.58% | 42.98% | 13.75% |
| Text-Only | GPT-4 | 78.0% | 86.86% | 82.86% |
| | GPT-4o | 80.0% | 83.14% | 88.57% |
| | Gemini-1.5-pro | 65.8% | 72.99% | 79.02% |
| | Claude-opus-3 | **80.57%** | 85.43% | 87.43% |
| | Claude-sonnet-3.5 | 79.43% | 82.29% | 89.43% |
| | Bart L mnli | 68.0% | 68.0% | 68.0% |
| | DeBeRTa-v3 | 65% | 65% | 65% |

Table 6: Pairs-setup performance with additional rank information regarding rank differences and absolute values.

| Strategy | Model | Accuracy (%) | ‖Rank Diff‖ | Equal Rank Rate (%) | Correct Rank Diff | Incorrect Rank Diff | Correct Rank | Incorrect Rank |
|---|---|---|---|---|---|---|---|---|
| Separate-Images | GPT-4o | 50 | 1.48 | 37 | 2.54 | 0.42 | 7.39 | 6.61 |
| | GPT-4-vision | 40 | 1.60 | 44 | 3.12 | 0.58 | 6.97 | 6.04 |
| | Gemini-1.5-pro | 42 | 1.71 | 43 | 3.16 | 0.65 | 7.13 | 6.47 |
| | Claude-Sonnet-3.5 | 38 | 1.33 | 41 | 2.49 | 0.60 | 7.97 | 7.58 |
| Combined-Image | GPT-4o | 45 | 1.30 | 37 | 2.27 | 0.50 | 7.77 | 7.23 |
| | GPT-4-vision | 34 | 1.28 | 48 | 2.52 | 0.62 | 6.76 | 6.37 |
| | Gemini-1.5-pro | 39 | 1.73 | 45 | 3.29 | 0.71 | 6.99 | 6.47 |
| | LLaVA 1.6 | 42 | 2.38 | 27 | 3.31 | 1.69 | 6.10 | 5.02 |
| | Fuyu-8b | 44 | 2.58 | 17 | 3.25 | 2.04 | 8.06 | 7.60 |

Table 7: Human votes of candidate explanations. The percentage of votes reflects annotators' agreement with the candidate explanations provided by the models. 0-votes notes no-selection, 3-votes notes selected unanimously.

| Input Strategy | Model Name | 0 Votes (%) | 1 Vote (%) | 2 Votes (%) | 3 Votes (%) |
|---|---|---|---|---|---|
| Humans | - | 5% | 9% | **25%** | **60%** |
| Separate-Images | GPT-4o | 77% | 8% | 5% | 9% |
| | GPT-4-vision | 61% | 8% | 6% | **25%** |
| | Gemini-1.5-pro | 62% | 9% | 7% | 23% |
| | Claude-sonnet-3.5 | 50% | 12% | 12% | **25%** |
| Combined-Image | GPT-4o | 56% | 6% | 8% | **31%** |
| | GPT-4-vision | 63% | 8% | 11% | 18% |
| | Gemini-1.5-pro | 60% | 7% | 12% | 21% |
| | Claude-sonnet-3.5 | 58% | 8% | 10% | 24% |
| | Claude-opus-3 | 74% | 12% | 5% | 9% |
| | LLaVA 1.6 | 85% | 9% | 5% | 1% |
| | Fuyu | 90% | 8% | 2% | 1% |

Table 8: Number of evaluated explanations. The explanations are associated with correct *plausibility prediction*. Human explanations include the correct explanations of 3 crowd-workers. The explanations are evaluated by human and automatically.

| Input Strategy | Model | Num Candidate Explanations |
|---|---|---|
| Humans | | 840 |
| Separate-Images | Gemini-1.5-pro | 282 |
| | GPT-4-vision | 278 |
| | GPT-4o | 340 |
| | Claude-sonnet-3.5 | 265 |
| Combined-Image | Claude-Opus-3 | 309 |
| | Claude-sonnet-3.5 | 276 |
| | LLaVA 1.6 | 224 |
| | Gemini-1.5-pro | 298 |
| | GPT-4-vision | 281 |
| | GPT-4o | 298 |
| | Fuyu-8b | 181 |
| Total Model explanations | | 3,868 |

GOLD
(TEXTUAL DESCRIPTIONS)

IMAGE-TO-TEXT
(CLAUDE 3.5)

IMAGE

a groom and bride hold hands in the aisle, looking happy.

'The image shows a **couple** in what appears to be a **church** or **chapel** setting. They are walking together and looking at each other with bright smiles, suggesting a joyful occasion, **possibly a wedding.**
The woman is wearing an elegant off-the-shoulder white dress with long sleeves, which could be a **wedding gown**. Her long blonde hair is styled in loose waves.
The man is dressed in a dark suit jacket, white shirt, and has a small floral boutonniere pinned to his lapel, which is typical for a groom. He has dark, wavy hair and a beard.
The background shows arched ceilings, hanging lights, and what looks like a stained glass window, all indicative of a church interior. There are hints of other people and decorations in the background, suggesting an event is taking place.
The overall atmosphere of the image is one of **happiness and celebration**, captured in an **intimate moment between the couple** as they walk through the venue.'

two phones (one belonging to a man, one to a woman) display the tinder app with a match.

'The image shows two smartphones held side by side, displaying dating app interfaces. **Both screens show "It\'s a Match!" messages**, indicating that **two users have matched with each other on the dating platforms**. The left phone displays an app called "Tindor" with a profile picture of a smiling man with dark hair and a beard. The right phone shows an app called "Tinde" with a profile picture of a laughing woman with long blonde hair. Both app interfaces feature similar layouts with heart icons, action buttons (likely for liking or rejecting profiles), and some text beneath the "It\'s a Match!" message. The phones are being held by hands, suggesting a person is holding each device. The overall tone of the image is warm, with a soft background lighting that emphasizes the bright screens of the smartphones. This setup appears to be illustrating the moment when **two people match on dating apps**, capturing the excitement and potential of a new connection in the digital dating world.'

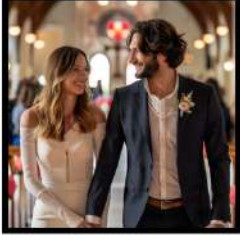

two phones (one belonging to a man, one to a woman) display the tinder app with a no-match.

'The image shows two smartphones side by side, displaying what appears to be a **dating app interface**. Both screens show profile pictures and the text "No Match" prominently displayed. On the left phone, we see a man with dark hair and a beard, looking down at his phone. On the right phone, there\'s a woman with long, light-colored hair, also looking at her phone while touching her face. Below the profile pictures on both phones, there are two buttons: a red heart on the left and a gray or black heart on the right. These are likely the app\'s interface for liking or passing on a potential match. The phones appear to be modern smartphones with notched screens, resembling recent iPhone models. The background has a warm, slightly glowing effect, creating a moody atmosphere. The overall composition suggests a commentary on **modern dating apps** and the potential **disconnection or missed connections** that can occur in digital matchmaking.'

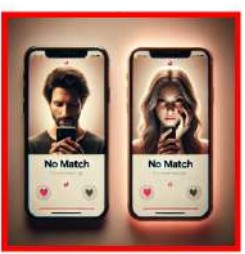

Figure 7: Image-to-text descriptions example. Detailed descriptions by Claude 3.5 and the gold textual descriptions. In bold style are the key necessary details for succeeding in the *plausibility prediction* task.

## C.2 FURTHER EXPERIMENTS

To complement our primary experiments, we conducted additional analyses to further examine the reasoning capabilities of Vision-Language Models (VLMs) under various conditions. These experiments focus on three key aspects: (1) evaluating the performance of additional open-source VLMs that support multi-image inputs and flexible resolutions (Table 9), (2) assessing image-to-text pipelines using Claude 3.5 as a predictor, in contrast to GPT-4o in the main experiments (Table 10), and (3) analyzing the sensitivity of VLMs to different prompt formulations (Table 11). Our findings highlight persistent limitations in current VLMs, with most models struggling to perform well on NL-EYE tasks.

Table 9: **Additional Open-source VLMs**: Scores for vision-based experiments, including two additional VLMs that are multi-image and support flexible resolutions and aspect ratios: MiniCPM, which employs a SigLIP-400M visual encoder paired with the MiniCPM-2.4B language backbone (MiniCPM v2.6), and LLaVA-OneVision, which integrates a SigLIP vision encoder with a Qwen2 language backbone (LLaVA-OneVision-Qwen2-7b). While performance on separate images remains below random, using a combined image strategy with these models demonstrates improvement, showcasing their enhanced ability to encode and leverage visual information effectively.

| Input Strategy | Model | Triplet Acc. |
|---|---|---|
| **Humans** | — | 85% |
| Separate Images | MiniCPM-V-2.6 | 12% |
| | LLaVA-OneVision-Qwen2-7b-ov | 18% |
| Combined Image | MiniCPM-V-2.6 | 36% |
| | LLaVA-OneVision-Qwen2-7b-ov | 23% |
| | LLaVA-1.6 | 14% |
| **Random** | — | 25% |
| **Dumb Pixel** | — | 50% |

Table 10: **Image-to-Text:** Performance drops were observed also with Claude 3.5 as a predictor, evaluated using Triplet Accuracy based on a separate image input strategy. GPT-4o demonstrates lower performance in the *plausibility performance* task, as it relies on image descriptions. Both models exhibit a significant drop in performance overall.

| Describer | Predictor: Claude 3.5 |
|---|---|
| Gemini-1.5-Pro | 50% |
| GPT-4 vision | 44% |
| LLaVA 1.6 | 36% |
| BLIP 2 | 42% |
| Instruct BLIP | 36% |

Table 11: **Prompts Comparison with GPT-4 Vision:** Analyzing different prompts provides valuable insights; however, a well-performing VLM (or any model) should be able to understand and follow instructions. Excessive sensitivity to input prompts may indicate difficulty with the task. This table includes results from experiments with 3 additional prompts. The CoT approach shows a 3% improvement, while overall, the prompts demonstrate comparable performance.

| Prompt Variant | Prompt Template | Triplet Acc. (Separated) |
|---|---|---|
| Regular (Table 16) | – | 46% |
| Reverse Task | *First explain, then predict* | 43% |
| CoT | *Let's think step by step* | 49% |
| Role | *You are a causality expert* | 44% |

## D    FAILURE ANALYSIS

This section examines key failure points and differences in VLM reasoning patterns, with a focus on quantitative comparisons to human performance and qualitative failure analysis.

**Are Human and Model Difficulties Aligned?**    In the *plausibility prediction* task using the triplet setup of the separate-images input strategy, humans and VLMs agree on 55% of the predictions (full comparison in Table 12). When humans are incorrect, the model's success rate falls to 30%, below random chance. While the model outperforms humans in only 5% of cases, its explanations in these instances are rarely accurate. Furthermore, only 21% of the models' errors overlap with human errors, indicating that humans and models tend to make different types of mistakes. Additional examples are shown in Figures 11 and 12.

Table 12: *Plausibility prediction* analysis: Model vs. Human Comparison. ✓notes correct prediction while × notes incorrect one.

| Model | Model ✓Human ✓ | Model ✓Human × | Model ×Human × | Model ×Human ✓ |
|---|---|---|---|---|
| Claude-sonnet-3.5 | 44% | 5% | 10% | 40% |
| Gemini-1.5-pro | 46% | 5% | 11% | 38% |
| GPT-4-vision | 41% | 4% | 11% | 42% |
| Models Avg. | 44% | 5% | 11% | 40% |

**VLM Failure Analysis**    This section presents a qualitative analysis of VLMs' explanations for wrong predictions. We qualitatively analyze 120 explanations – 40 from each VLM: Gemini 1.5, GPT-4 and Claude 3.5.

We identify five main factors contributing to the models' failures: (1) *style & consistency*: When irrelevant visual details influence the decision; (2) *time*: When the explanation relies on incorrect time direction or duration; (3) *ignoring key details*: Overlooking important information; (4) *missing knowledge*: Misinterpreting key details despite recognizing them; (5) *failed comparison*: Justifying a less plausible hypothesis with logical reasoning. Table 13 presents illustrative examples of these factors.

As Figure 8 shows, all models struggle with understanding temporal progression. Notably, Claude often relies on style considerations, with 30% of its errors resulting from this factor, indicating an overemphasis on irrelevant visual details. Both Gemini (32%) and GPT-4 (25%) frequently miss key details, suggesting recognition gaps. GPT-4 has the highest rate of failed comparisons, often making the incorrect decision at the final plausibility stage. To further understand these failures, researchers can try to interpret models' internal thought processes (Toker et al., 2024).

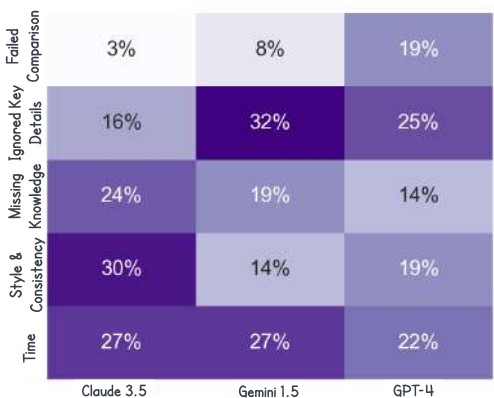

Figure 8: Failure factors of model explanation for incorrect plausibility prediction.

**Does Image Naturalness Influence Model Performance?**    Synthesizing the benchmark provides flexibility to simulate diverse everyday scenes while maintaining consistency and quality. Extracting triplet scenes from real-world sources, such as videos, poses challenges in terms of efficiency and feasibility, particularly when ensuring consistency between the premise and a false hypothesis, which may not exist in the same video.

To evaluate the influence of image style, generated versus realistic, on model performance, we conducted an ablation study using a subset of 20 triplets (60 images) [3] that are parallel to the generated

---

[3]The sample IDs of the real images subset: [8, 15, 24, 30, 35, 47, 51, 63, 84, 86, 106, 118, 187, 192, 200, 256, 268, 312, 342]

Table 13: Failure factors with examples, as illustrated by the following scene - a premise of *a man in a hospital bed with a broken leg* and two hypotheses: *a wet floor with (less plausible) and without (plausible) a warning sign* (Figure 1).

| Failure Factor | Example |
|---|---|
| *Style & Consistency* | Selecting the image because the window in the background matches the one in the premise image. |
| *Time* | Assuming the hypothesis occurs after the premise and concluding the warning sign was placed after an accident. |
| *Ignored Key Details* | Ignoring the cast on the man's leg and assuming he's resting rather than injured. |
| *Missing Knowledge* | Identifying the sign but not realizing it's a warning sign. |
| *Failed Comparison* | Selecting the hypothesis with the warning sign, reasoning the person was distracted and didn't see it, even though slipping could occur in both cases. |

ones, consisting solely of natural images sourced online or photographed by students, as demonstrated in Figure 9. We selected examples that were straightforward to produce, focusing on those that did not require consistency across individuals or highly nuanced setups, due to the complexity of finding or producing such cases. This resulted in simpler examples that generally yielded higher performance outcomes.

Our analysis reveals that GPT-4 Vision achieves 58% consistency-accuracy (triplet, separate images) on real images compared to 68% on generated original ones. This suggests that the performance gap is not significantly influenced by the type of images used, but it opens the door for further investigation into how visual styles may may impact the robustness of visual abductive reasoning performance in VLMs.

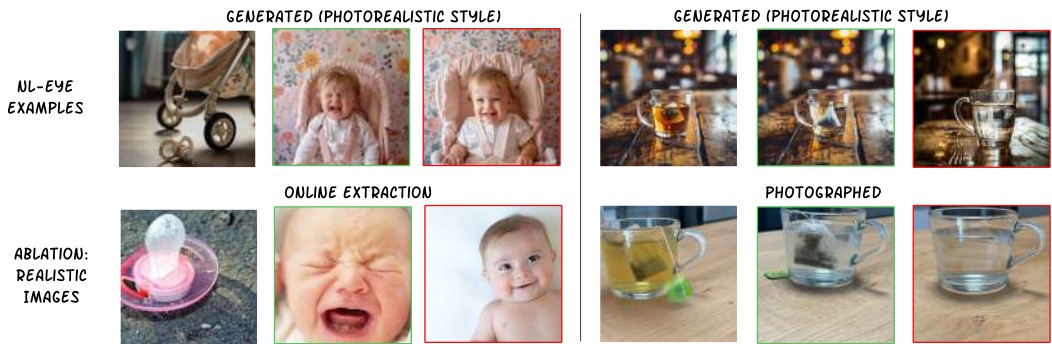

Figure 9: Examples of the real images subset.

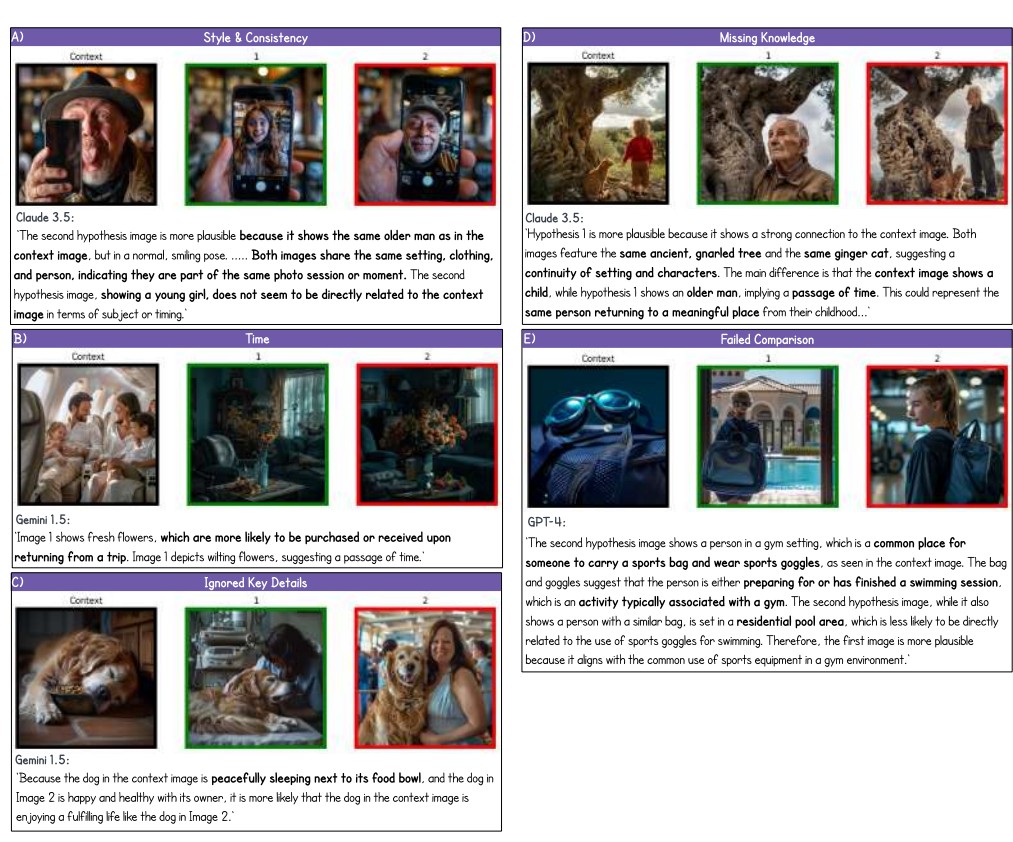

Figure 10: VLM failure analysis: Explanations examples. Based on five main factors: (A) Style & consistency, (B) Time, (C) Ignored key details, (D) Missing knowledge and (E) Failed comparison.

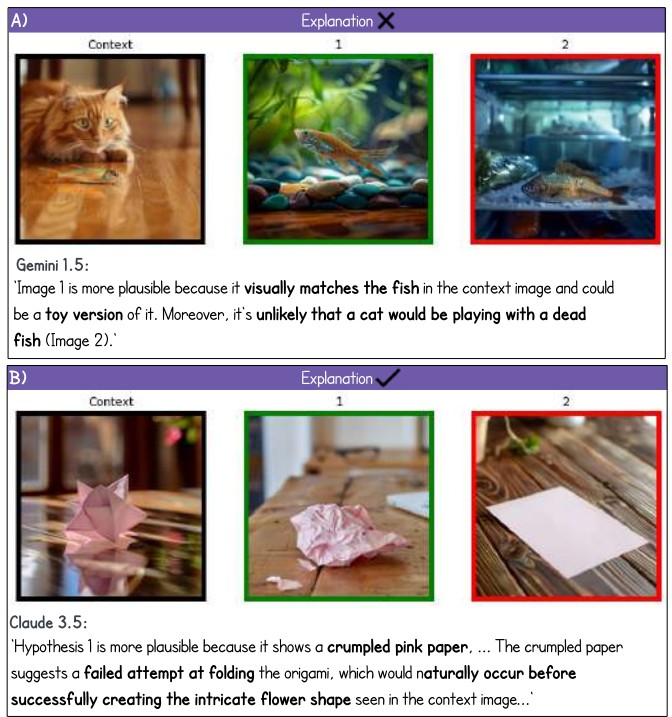

Figure 11: VLM failure analysis: When the model's plausibility prediction is correct - the explanation can be either valid (B) or not (A).

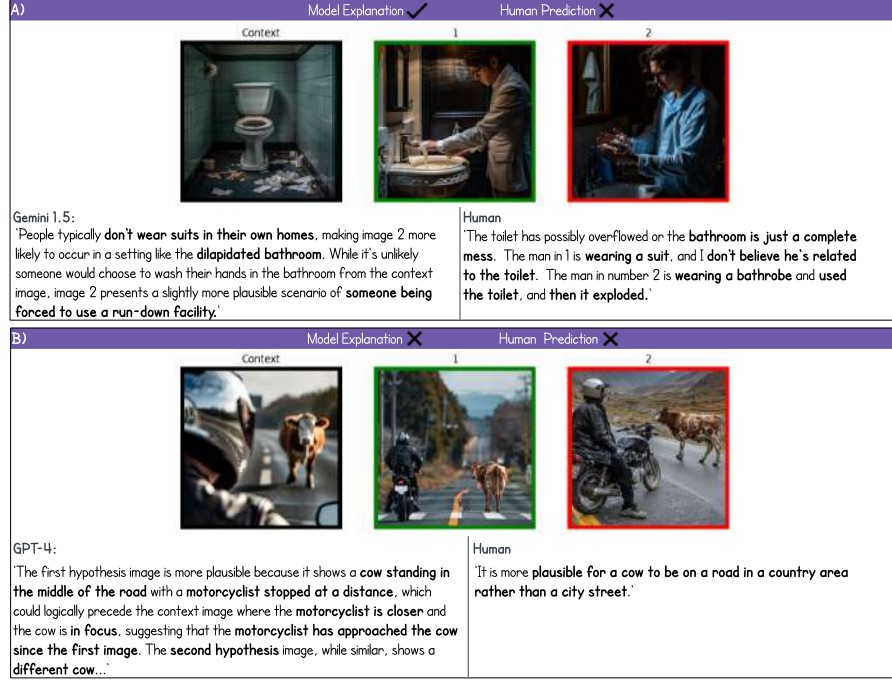

Figure 12: VLM failure analysis: When humans plausibility prediction is incorrect, and model's explanation is correct - the explanation can be either valid (B) or not (A).

# E   HUMAN PERFORMANCE & EVALUATION

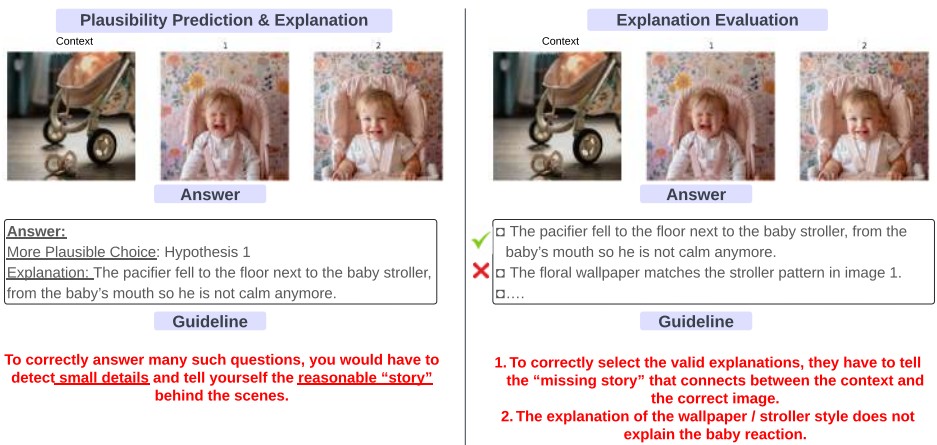

Figure 13: Guidelines for the crowd-workers. Guidelines for the human baseline on the plausibility prediction and plausibility explanation tasks (left), and for human evaluation of explanations (right).

**Human Performance - Plausibility Prediction and Explanation.**   AMT Crowd-workers were instructed to complete the plausibility prediction and explanation tasks based on the following guidelines, in Figure 13 and the questions in Figure 14.

**Human Evaluation of Explanations.**   The explanations are presented in a multiple-choice question format (see Figure 15), where the crowd workers are instructed to select explanations that demonstrate logical reasoning and clearly justify why the correct hypothesis is more plausible than the other candidate (see Figure 13). We conduct the human evaluation of explanations across all the input formats in the vision-based reasoning approach (Triplet setup), focusing on the explanations associated with the correct plausibility predictions, resulting in a total of 3.8k explanations.

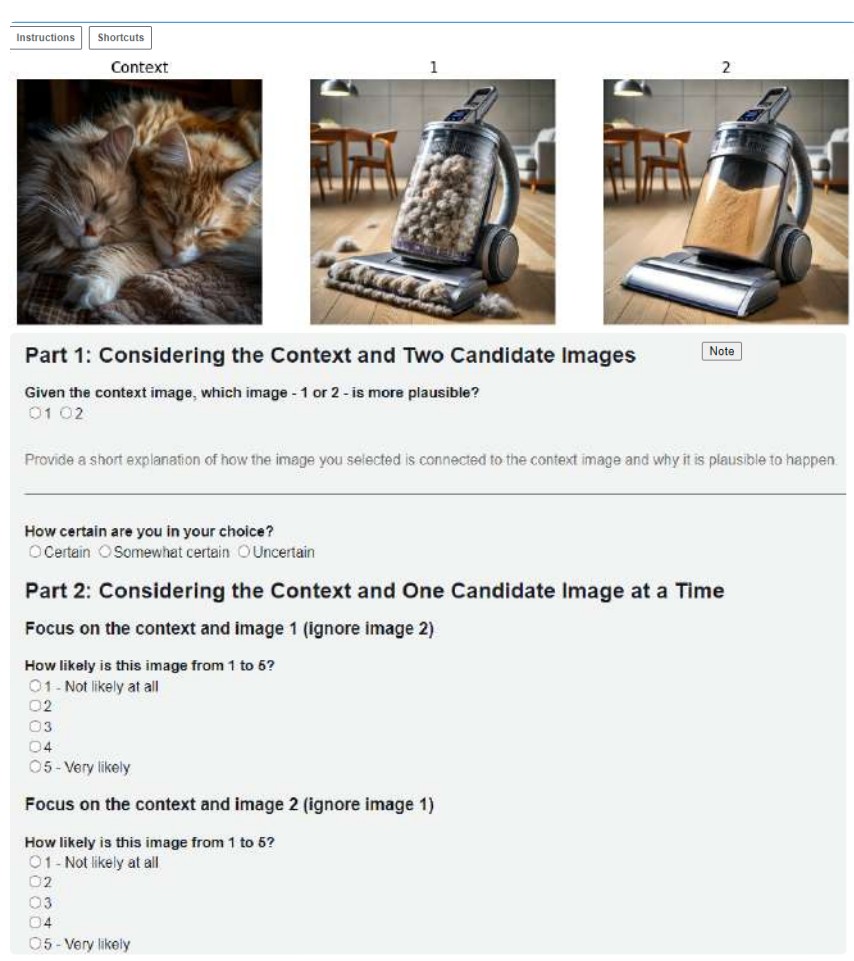

Figure 14: AMT questionnaire (*human baseline*) screen of the plausibility prediction and plausibility explanation tasks.

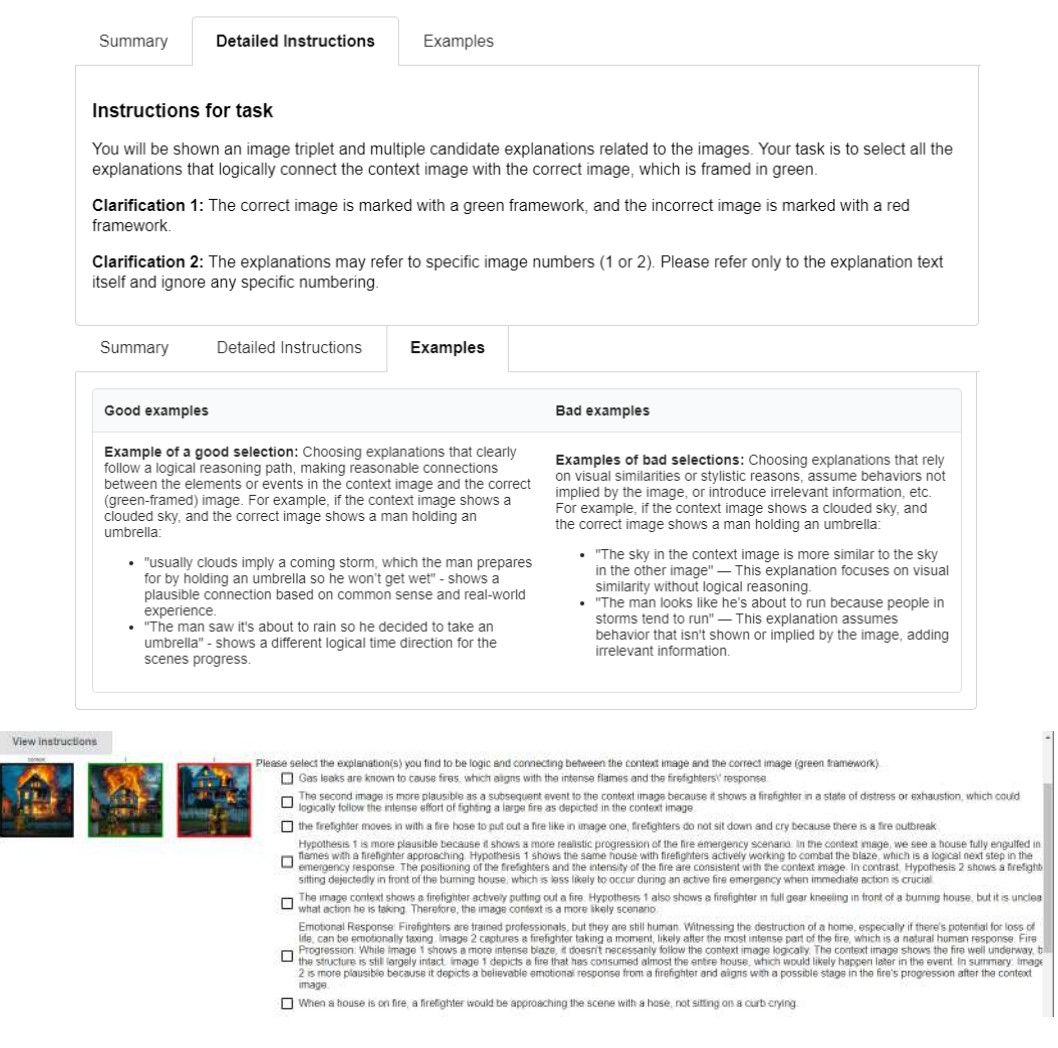

Figure 15: Human evaluation of explanation screen, including (a) instructions provided to participants, and (b) a screenshot of the AMT questionnaire.

# F    DISCUSSION AND FUTURE DIRECTIONS FOR ENHANCING VISUAL ABDUCTIVE REASONING

To improve VLMs' abductive reasoning, future efforts could focus on refining both model architectures and training tasks. Given our findings that models can reason from textual descriptions but struggle with images, one approach might be to modify the image encoding to align with the textual query, thereby enhancing representations of specific areas within the image. Another potential solution is to train a model to generate image descriptions that are optimized for downstream reasoning tasks.

Building on this, techniques like Chain-of-Thought (CoT) can be extended to the visual domain, generating images as part of the reasoning-thinking process. Additionally, VLMs tend to assign undue importance to style or image order rather than focusing on semantics. To address this, constructing a large-scale training set that emphasizes semantic content over order and style could help models prioritize meaning over superficial patterns, resulting in more robust reasoning. These strategies collectively offer a path toward enhancing models' visual abductive reasoning capabilities.

# G    NL-EYE EXAMPLES

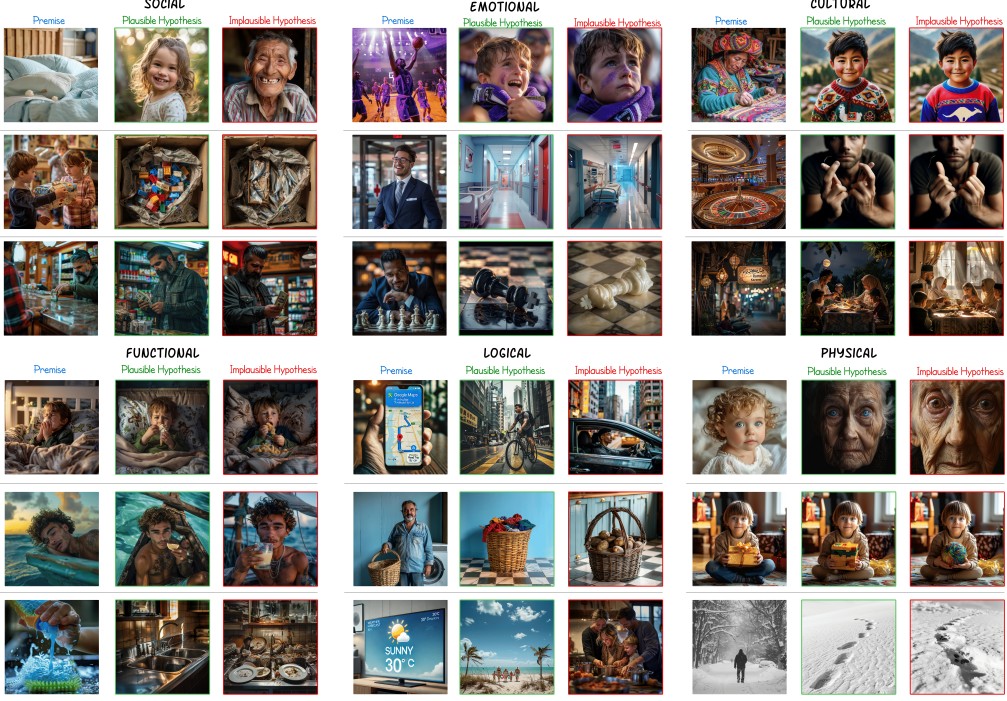

Figure 16: NL-EYE examples from each reasoning category (3 triplets per category). Each example consists of 3 images: a premise (left column), a plausible hypothesis (green frame), and a less plausible hypothesis (red frame). While the gold explanations are included in the benchmark, we invite the reader to attempt to create valid explanations on their own.

## H    NL-EYE DATASET: CURATION DETAILS

Table 14: Examples of suggested textual descriptions (scenes) filtered by specific criteria.

| Filter Criterion | Premise | Hypothesis 1 | Hypothesis 2 | Reason |
|---|---|---|---|---|
| Premise necessity | *A teacher enters school* | *An apple on the teacher's desk* | *An orange on the teacher's desk* | We don't need to see the teacher to understand it's a school setup |
| Visual relevance | *Man says hello* | *Man enters home* | *Man exits home* | It's unclear if the man is saying hi or bye |
| Uniqueness | *Man with a broken leg* | *Hole in a road under construction* | *Hole in a road with a warning sign of completed work* | Repetition of existing ideas (Figure 1) |

**Text-to-Image Prompt.**    The Text-To-Image prompt (in Midjourney) is consisted of 3 parts, while the last one is optional:

- **Text description.** The textual scene caption, basic or improved.
- **Photorealistic style.** Adding textual styling of photorealistic images by mentioning it is captured with Nikon D850.
- **Visual consistency.** Making an image consistent with another image by setting the same seed number, and referring to the reference image with the flag *cref* and its conditioning strength with the flag *cw* ranging from 0-100.

All these parts are aggregated into the following template:

*< prompt caption >, captured with a Nikon D850 and a 24-70mm lens at f/2.8 –seed <> –cref <> –cw 80*

**Prompt Augmentation.**    Augmenting a text description by prompting Gemini or GPT-4 with the following prompt:

*Describe visually a specific looks of < interacting component1 >, < interacting component2 > and < environment >. keep it short and concise, and avoid NSFW words. and integrate these details into every reference of them in the following captions smoothly and consistently. do not change the content of the captions besides the visual description integrations. return in a JSON format: 1) <first image caption> 2) <plausible second image caption> 3) <implausible second image caption>*

*Note: integrate the environment only if it fits the context of the caption.*
*For example:*
*interacting component1: little child, interacting component2: vaccine, environment: nurse room, first image caption: a child gets a vaccine., plausible second image caption: a child cries after getting a vaccine., implausible second image caption: a child smiles after getting a vaccine.*
*Response:*
*improved first image caption: a short curly-haired child wearing a green t-shirt receives a vaccine with a silver syringe in a nurse's room filled with toys. improved plausible second image caption: the short curly-haired child in a green t-shirt cries after receiving a vaccine in the toy-filled nurse room. improved implausible second image caption: the short curly-haired child in a green t-shirt smiles after receiving a vaccine in the nurse's room.*

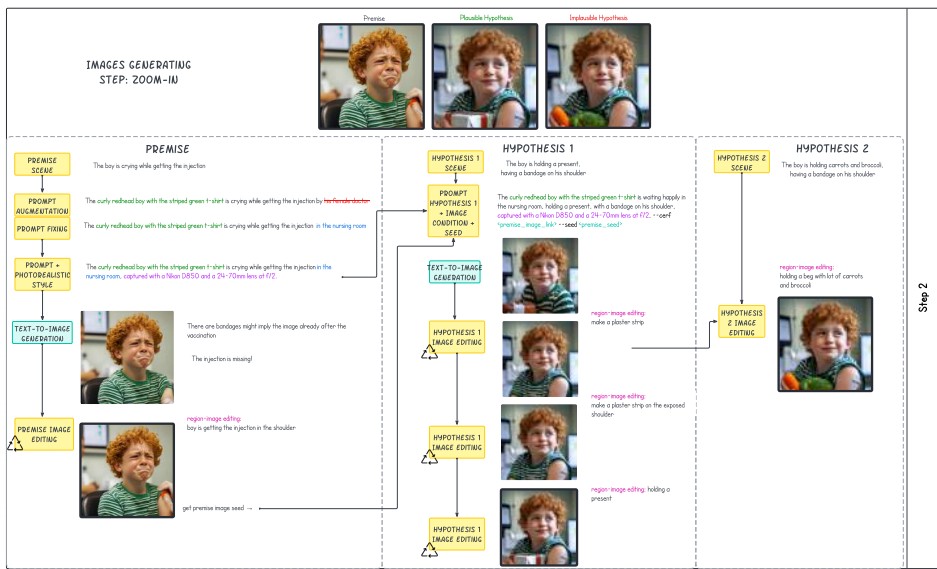

Figure 17: Zoom into the image generation step in NL-EYE curation, as seen in Figure 6. Yellow color notes a hand-curated stage, while turquoise notes a model-generated stage. All stages require human involvement for fixing, editing and validating.

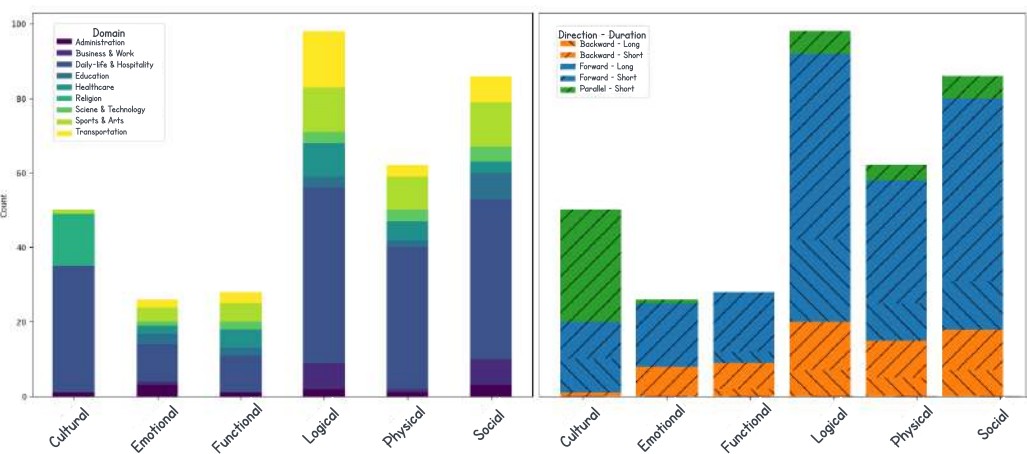

Figure 18: Dataset Analysis. A histogram of the NL-EYE examples. The benchmark is also annotated with diverse domains (left): *administration, business & work, daily life & hospitality, education, healthcare, religion, science & technology, sports & arts and transportation*, and representation of time duration and direction (right) in every reasoning category. Parallel in noted by "parallel-short".

## I  REPRODUCIBILITY AND RESOURCES

Table 15: API and version of closed-source models used for inference on NL-EYE tasks.

| API | Model Version | Used as |
|---|---|---|
| Gemini | gemini-1.5-pro | VLM, LLM |
| GPT | gpt-4o-2024-08-06 | VLM, LLM |
| GPT | gpt-4-1106-vision-preview | VLM, LLM |
| Claude | claude-3-opus-20240229 | VLM, LLM |
| Claude | claude-3-5-sonnet-20240620 | VLM, LLM |

Table 16: Textual prompts for task descriptions in different input strategies and setups.

| Input Strategy | Setup | Prompt Template |
|---|---|---|
| Separate-Images | Triplet | *Given a context image and 2 hypothesis images (3 total images), which image of the following two (1 and 2) is more plausible? The context image can happen before or after the hypothesis images. Mention which one is more plausible – 1 or 2, and explain.* |
|  | Pairs | *Given a pair of images – a context image and a hypothesis image – rank how plausible the hypothesis image is in relation to the context. The context image can occur before or after the hypothesis image. Rank the plausibility with a score between 1 and 10, where: 1: Not plausible at all, 3: Slightly plausible, 5: Moderately plausible, 7: Very plausible, 10: Almost necessarily plausible. Explain why.* |
| Combined-Image | Triplet | *Given a context image (left image) and two hypothesis images (middle and right), which hypothesis image (1 or 2) is more plausible? Mention which one is more plausible – 1 or 2, and explain. The context image can happen before or after the hypothesis images.* |
|  | Pairs | *The first (left) image is the context image. Given a pair of images...(as Separate-Images - Pairs)* |
| Text-Only | Triplet | *Given a context, hypothesis1, and hypothesis2, which hypothesis is more plausible? The context can occur before or after the hypotheses.* |
| Image-to-Text | Triplet | *Describe the content of the image in detail.* |

**Prompt of Auto Evaluation of Explanation.** We combine classes 0 and 1 as a false (invalid) explanation, and 3 as a positive (valid) explanation:

*Context: <premise textual description>,*
*Plausible Hypothesis: <hypthesis 1 textual description>,*
*Less Plausible Hypothesis: <hypothesis 2 textual description>.*
*Return 0 (Not logical at all), 1 (Logical but different), or 2 (Logical and same as one of the gold)*
*if the candidate's explanation presents the same logical common sense as appears in one of the gold explanations for justifying the plausible hypothesis.*
*Candidate explanation: <candidate explanation>.*
*Gold explanations:*
*Explanation 1: <gold explanation>...*

