# OpenReview forum: "NL-Eye: Abductive NLI For Images"
_ICLR.cc/2025/Conference — ICLR 2025 Poster_

### Official Review · Reviewer_972W · 2024-11-03

**Soundness:** 4
**Presentation:** 3
**Contribution:** 3
**Rating:** 6
**Confidence:** 3

**Summary:**

This paper introduces NL-EYE, a benchmark designed to test the abductive reasoning capabilities of Visual Language Models (VLMs) through image-based tasks. The benchmark includes 350 carefully curated triplet examples spanning diverse reasoning categories where models must choose the more plausible hypothesis from a set and provide an explanation. Experiments reveal that while humans excel in this task, current VLMs show notable deficiencies in their reasoning capabilities. The authors conclude that VLMs face significant challenges in visual interpretation, which impacts their ability to reason effectively about images.

**Strengths:**

1. The benchmark is well-designed with diverse reasoning categories.
2. Experiments on the benchmark reveal interesting findings.
3. The analysis is thorough and highlights notable insights into VLM limitations.

**Weaknesses:**

1. While I agree that the benchmark is carefully curated, the filtering condition can be inconsistent and subjective because it is done manually.
2. This paper focuses primarily on evaluating VLMs' deficiencies but lacks discussion on strategies or methods to improve these models' abductive reasoning capabilities.
3. The paper lacks experiments with additional open-source models. While the current model selection is valid, given the paper's findings about failures in visual interpretation and hypothesis location dependency, testing VLMs with different visual encoders or those trained on multi-image datasets would further support the analysis.

**Questions:**

## Question
1. Have the authors conducted experiments with VLMs that trained on datasets including multiple images such as LLaVA-Onevision or VILA, and with VLMs that use other visual encoders like Cambrian-1?

## Typo
* L260 Validation,and Categorization -> Validation and Categorization

---
### References
* Lin, Ji, et al. Vila: On pre-training for visual language models. CVPR 2024
* Li, Bo, et al. Llava-onevision: Easy visual task transfer. https://llava-vl.github.io/blog/2024-08-05-llava-onevision/
* Tong, Shengbang, et al. Cambrian-1: A fully open, vision-centric exploration of multimodal llms. Neurips 2024

---

> ### Author Response · Authors · 2024-11-18
>
> Thank you for your thoughtful feedback on the paper. We greatly appreciate your recognition of its strengths and the valuable suggestions provided. We will make an effort to address these points to further enhance the clarity and impact of our work.
>
> **Filtering criteria**
>
> Thank you for highlighting the filtering step in our benchmark curation process. We intentionally applied manual filtering, as it was crucial for ensuring the highest quality and could not be effectively achieved through automated methods. Manual curation allowed us to apply careful judgment, ensuring that: (1) the premise was indispensable for predicting the hypothesis, (2) examples were visually expressible, and (3) novel ideas were prioritized to enhance the diversity of the benchmark. These criteria, along with illustrative examples, are detailed in Table 13 of our paper.
>
> **Strategies discussion**
>
> Thank you for this suggestion. Exploring ways to enhance reasoning capabilities in VLMs is indeed an interesting direction. In the paper, we included a subsection and analysis discussing possible reasons for model failures (L520-L523), highlighting several areas for future improvement and suggesting specific interventions. We emphasize in Section 5.2 that while the models demonstrate strong textual reasoning, visual reasoning remains a challenge and thus represents a promising area for improvement. Additionally, we discuss aspects such as image order (L493-L501) and reasoning types (L503-L514) that pose particular difficulties for the models. Our analysis (Section 6) identifies five key factors contributing to incorrect predictions, such as style inconsistencies and failed comparisons during "last-minute" decision-making. These factors lay the groundwork for a deeper exploration of strategies to improve visual reasoning capabilities.  That said, it is important to note that the primary focus of our paper is to introduce a new task with a high-quality benchmark and to assess the performance of state-of-the-art vision-language models (VLMs) on this task.
>
> In light of your suggestion, we have added a discussion to the paper on future efforts, focusing on refining image-to-text alignment, optimizing descriptions, applying visual Chain-of-Thought techniques, and prioritizing semantics over style and order in training to enhance abductive reasoning.
>
>
>
> **Additional open-source models**
>
> Following your suggestion, we incorporated two additional open-source VLMs into our analysis, both utilizing the SigLIP visual encoder (in contrast to the CLIP-based encoder used in LLaVA 1.6) with distinct language backbones:
>
>
> - **MiniCPM v2.6**: Leverages a SigLIP-400M visual encoder paired with the MiniCPM-2.4B language backbone.
> - **LLaVA-OneVision-Qwen2**: Integrates a SigLIP visual encoder with the Qwen2-7B language backbone.
>
>
> Our findings indicate that while performance on separate images remains below random, adopting a combined image strategy with these models resulted in noticeable improvements. This demonstrates that the SigLIP visual encoder enhances the models' ability to encode and utilize visual information effectively. These insights and experiments will be incorporated into the paper.
>
>
> | Input Strategy    | Model            | Triplet Acc. (%) |
> |--------------------|------------------|-------------------|
> | Humans            | Humans           | 85%              |
> | Separate Images    | MiniCPM          | 12%              |
> | Separate Images    | LLava-onevision  | 18%              |
> | Combined Image     | MiniCPM 2.6      | 36%              |
> | Combined Image     | LLava-onevision  | 23%              |
> | Combined Image     | LLava-1.6        | 14%              |
> | Baselines          | Random           | 25%              |
> | Baselines          | Dumb Pixel       | 50%              |

---

> > ### Comment · Reviewer_972W · 2024-11-28
> >
> > Thank you for your detailed response and the additional experiments with SigLIP-based models. The results provide valuable insights into the impact of visual encoders on model performance. I will increase Soundness to 4.
> >
> > Could you include results for DINOv2-based models such as Cambrian-1? Comparing their results against the current models would help understand how different pre-training approaches affect abductive reasoning capabilities.

---

> > > ### Author Response · Authors · 2024-12-01
> > >
> > > Thank you for appreciating the new experiments and the increased soundness score. Below are the results, including Cambrian-1 (DINOv2-based model, 8b), evaluated under the combined image input strategy, comparing the impact of different image encoders on abductive reasoning.
> > >
> > > | Input Strategy  | Model           | Image Encoder             | Triplet Acc. (%) |
> > > |----------------------|---------------------|--------------------------------|-----------------------|
> > > | Humans              | Humans             | -                              | 85%              |
> > > | Separate Images     | MiniCPM            | SigLIP                         | 12%                  |
> > > | Separate Images     | LLava-onevision    | SigLIP                         | 18%                  |
> > > | Combined Image      | MiniCPM 2.6        | SigLIP                         | 36%                  |
> > > | Combined Image      | LLaVA-onevision    | SigLIP                         | 23%                  |
> > > | **Combined Image**      | **Cambrian-1**         | **SVA (CLIP, SigLIP, DINOv2)**     | **19%**               |
> > > | Combined Image      | LLaVA-1.6          | CLIP                           | 14%                  |
> > > | Baselines           | Random             | -                              | 25%                  |
> > > | Baselines           | Dumb Pixel         | -                              | 50%                  |
> > >
> > > Note: SVA refers to the Spatial Vision Aggregator [1].
> > >
> > > ---
> > >
> > > [1] **Cambrian**: [https://cambrian-mllm.github.io/](https://cambrian-mllm.github.io/)
> > >
> > > We believe this addition strengthens our work and kindly ask you to reconsider the score to improve its chances of acceptance. Thank you again for your thoughtful review.

---

### Official Review · Reviewer_GcX5 · 2024-11-03

**Soundness:** 2
**Presentation:** 3
**Contribution:** 3
**Rating:** 5
**Confidence:** 4

**Summary:**

This work proposes the NL-Eye benchmark to evaluate the abductive reasoning ability of visual-language models from pure visual perception in multi-image situations, inspired by Natural Language Inference (NLI). The benchmark consists of testing examples for temporal reasoning along with six reasoning categories and images are obtained from the text-to-image generation model. The authors argue that current visual-language models show significantly inferior performance compared to humans on abductive reasoning in multi-image situations and claim that this is due to the lack of purely visual perception abilities compressed from the visual perception modules.

**Strengths:**

This work is distinct from existing multi-image benchmarks in that the objects of perception required to perform reasoning are provided solely through visual perception. NLI-inspired benchmarks that require visual reasoning over multi-images already exist, such as [1], but they are limited in terms of evaluating purely on visual perception, as they require reasoning over a given natural language premise. However, NL-Eye has the unique feature that requires reasoning on pure visual perception, since these premises are provided as images.

[1] A Corpus for Reasoning About Natural Language Grounded in Photographs (Suhr et al., 2019)

**Weaknesses:**

There is a lack of consideration in the experiments as to whether a proper evaluation of the current visual-language model can be made in a multi-image setting. As the authors argue, current benchmarks for testing abductive reasoning are single-image focused, but it should not be overlooked that research on the visual-language model itself is also focused on this. As a result, the authors provide “concatenated” images, which may not be a fair assessment for most visual-language models that currently operate at fixed, squared-sized resolutions. To demonstrate the need for the proposed benchmark, it is required to observe if the same phenomenon is found in visual-language models that can handle flexible resolutions and aspect ratios like [1].

[1] LLaVA-UHD: an LMM Perceiving any Aspect Ratio and High-Resolution Images (Guo et al., 2024)

**Questions:**

It would be nice to be able to determine if the problem this benchmark shows is out-of-domain on the language model side or a limitation of the visual encoder itself. If we split the data in the benchmark for training and test purposes and fine-tuned models improved on the remaining test splits, then we can assume that the main problem was the task was out-of-distribution rather than a lack of performance on visual perception, since most current visual-language models trained with a frozen visual encoder. Have you done any further experiments to see if this limitation on visual reasoning can be improved with some training or not?

---

> ### Author Response · Authors · 2024-11-18
>
> Thank you for your valuable feedback on the paper. We appreciate your perspective and suggestion and we will try to address these points to enhance the clarity of our work.
>
>
> **Multi-image setting**
>
> Thank you for raising this valuable point. We would like to highlight that all the examined state-of-the-art models—Gemini, GPT-4-Vision, and Claude—explicitly declare their support for multiple images [4,5,6].
>
> You raise a valid point about multi-image handling, which we addressed in the paper by implementing two input strategies: separate and combined images (Section 4.1, L317-328). Our results indicate performance degradation with the combined strategy for most models, likely due to challenges in encoding dense information into a single image.
>
> To further address this issue, we incorporated two additional VLMs that are multi-image and support flexible resolutions and aspect ratios:
>
> - **MiniCPM [1,2]**: This model utilizes the same technique as LLaVA-UHD, leveraging a SigLIP-400M visual encoder paired with the MiniCPM-2.4B language backbone. (MiniCPM v2.6)
> - **Llava-Onevision [2]**: This model integrates a SigLIP vision encoder with a Qwen2 language backbone. (LLava-onevision-qwen2-7b)
>
>
> While the performance on separate images remains below random, employing a combined image strategy with these models has shown improvement, highlighting their enhanced ability to encode and utilize visual information more effectively. This suggests that the proposed approach is both a valuable and valid contribution, and we are pleased to incorporate it into the paper.
>
> | Input Strategy    | Model            | Triplet Acc. (%) |
> |--------------------|------------------|-------------------|
> | Humans            | Humans           | 85%              |
> | Separate Images    | MiniCPM          | 12%              |
> | Separate Images    | LLava-onevision  | 18%              |
> | Combined Image     | MiniCPM 2.6      | 36%              |
> | Combined Image     | LLava-onevision  | 23%              |
> | Combined Image     | LLava-1.6        | 14%              |
> | Baselines          | Random           | 25%              |
> | Baselines          | Dumb Pixel       | 50%              |
>
>
>
> **Training**
>
> Thank you for this question. The NL-Eye benchmark is designed strictly as a test set, containing 350 carefully curated examples with human involvement at each stage. Due to its selective curation and small size, it is not intended for fine-tuning.
>
> To differentiate between limitations in visual perception and language comprehension, we used separate vision-based and text-based reasoning approaches. Our results indicate that VLMs perform better in text-based reasoning, while their challenges primarily lie in visual interpretation, as evidenced by higher performance in text-based tasks over vision-based ones. While NL-Eye itself isn’t suited for fine-tuning, we agree that further fine-tuning on a larger external dataset could offer valuable insights into potential improvements in both visual and textual reasoning. It is a great idea for future work 🙂
>
>
> [1] Yao, Y., Yu, T., Zhang, A., Wang, C., Cui, J., Zhu, H., ... & Sun, M. (2024). Minicpm-v: A gpt-4v level mllm on your phone. arXiv preprint arXiv:2408.01800.‏,
>
> [2] Hu, S., Tu, Y., Han, X., He, C., Cui, G., Long, X., ... & Sun, M. (2024). Minicpm: Unveiling the potential of small language models with scalable training strategies. arXiv preprint arXiv:2404.06395.‏
>
>
> [3] Li, B., Zhang, Y., Guo, D., Zhang, R., Li, F., Zhang, H., ... & Li, C. (2024). Llava-onevision: Easy visual task transfer. arXiv preprint arXiv:2408.03326
>
> [4] https://docs.anthropic.com/en/docs/build-with-claude/vision#example-multiple-images
>
>
> [5] https://ai.google.dev/gemini-api/docs/vision?lang=python#upload-local
>
>
> [6] https://platform.openai.com/docs/guides/vision#multiple-image-inputs

---

### Official Review · Reviewer_fepb · 2024-11-04

**Soundness:** 3
**Presentation:** 3
**Contribution:** 3
**Rating:** 6
**Confidence:** 4

**Summary:**

This paper presents NL-EYE, a benchmark to evaluate VLMs' visual abductive reasoning skills across six reasoning types: physical, functional, logical, emotional, cultural, and social. It includes 350 triplet examples (1,050 images) with temporal annotations indicating event sequence and duration. The study examines model robustness, considering hypothesis order sensitivity and different input formats (individual versus composite images). NL-EYE also assesses models' ability to score single hypotheses, addressing real-world scenarios where multiple alternatives may not be available.

**Strengths:**

1. This paper demonstrates a way to measure VLMs abductive reasoning ability on six diverse reasoning categories using multiple images.
2. The experiments shown in the paper comprehensively evaluate the reasoning ability of VLMs by checking image ordering, exploring different input formats to justify the reasoning gap of the existing VLMs.
3. The analysis section is interesting. The breakdown of performance across reasoning categories and the underlying insights will be useful for the community.

**Weaknesses:**

1. The prompt selection is under-explored.
2. More detailed in the Questions section

**Questions:**

Q1. It is unclear from the paper how the authors selected the concepts for each individual reasoning category. For example, in the Cultural Reasoning category, which cultures were represented in the generated image. As image generation models are also not good for cultural content generation and the VLMs being better on cultural NLI raise interest in which cultures were highlighted mostly in the data to assess the comprehensiveness of the test set.

Q2. The current prompt for VLM is asking the plausible answer first and then asking for explanation. It would be interesting to reverse this process (i.e., explain each image step-by-step and then conclude the plausible answer) and see how the VLMs react.

Q3. In Tables 2 and 3, LLaVA 1.6 performs better at predicting the plausible image using GPT-4 when converting image to text (Table 3) than when directly inputting images (Table 2). Could this difference be due to LLaVA’s limitations as a predictor, or is the prompt structure (e.g., asking for image descriptions first before selecting a plausible answer) affecting performance?

---

> ### Author Response · Authors · 2024-11-18
>
> Thank you for acknowledging the reasoning diversity in the benchmark, as well as the comprehensive evaluation, insights, and analysis. We will try to address each of your comments and questions in detail.
>
> **Prompts selection**
>
> In our study, the prompt was manually optimized using a small subset of examples. We deliberately chose a single, consistent prompt to maintain a controlled evaluation environment, focusing on performance differences between models rather than optimizing prompts for each model.
> While exploring alternative prompts can yield valuable insights, it’s crucial to note that a robust VLM (or any model) should effectively interpret and follow instructions. Excessive sensitivity to input prompts indicates potential task-related weaknesses. To investigate this, we are conducting experiments with three additional prompts.
>
> One of the additional prompts we are testing, inspired by your suggestion, is the "Reverse Task," which reorders the instructions within the prompt.
> Preliminary results are included below, and we plan to add this ablation study to the paper.
>
> Results on GPT-4 Vision:
>
>
> | Prompt Variant  | Prompt Template Change              | Triplet Acc. (Separated) (%) |
> |------------------|------------------------------|-------------------------------|
> | Regular          | --                           | 46%                          |
> | Reverse Task     | First explain, then predict  | 43%                          |
> | CoT              | Let’s think step by step     | 49%                          |
> | Role             | You are a causality expert   | 44%                          |
>
> **Concepts of reasoning categories**
>
> We agree that image generation models may fail to accurately generate cultural content [1]. This is why we choose to categorize the examples after the image generation phase. As discussed in lines 260-267, human annotators first validate the image-text alignment and then categorize the examples. We will make sure to mention the potential tendency of VLMs to fail to accurately generate cultural content and clarify how our methodology mitigates this.
>
> We found the following proportions: 8% American, 8% Jewish, 6% Japanese, 6% Indian, 6% Superstition, 6% Chinese, 4% Spanish, 4% Muslim, 4% Arab, 4% Hinduism, and 2% for each of the following: Amish, Asian, Mexican, Buddhist, Brazil, Western, Singaporean, Moroccan, Maori, Iranian, Scottish, Peruvian, Swiss, Medival, Swedish, and Russian.
>
> To clarify the distribution of cultures within the benchmark, we will update the following information in the paper as well.
>
>
>
> **LLaVA 1.6 performance**
>
> LLaVA v1.6 performs particularly well when tasked with describing the content of images, excelling in visual recognition and detection. However, its predictive capabilities, particularly in reasoning or decision-making tasks, are less developed compared to GPT-4. This limitation likely stems from its training, which has been primarily focused on Visual Question Answering (VQA) tasks. As such, LLaVA currently demonstrates greater strength as a descriptor than as a predictor [2].
>
>
> [1] Ventura, M., Ben-David, E., Korhonen, A., & Reichart, R. (2023). Navigating Cultural Chasms: Exploring and Unlocking the Cultural POV of Text-To-Image Models. arXiv preprint arXiv:2310.01929.‏
>
> [2] https://llava-vl.github.io/blog/2024-01-30-llava-next/

---

### Official Review · Reviewer_LDJN · 2024-11-04

**Soundness:** 3
**Presentation:** 3
**Contribution:** 3
**Rating:** 6
**Confidence:** 4

**Summary:**

This paper introduces a new benchmark NL-EYE that is designed to assess VLMs’ visual abductive reasoning skills. NL-EYE adapts the abductive Natural Language Inference (NLI) task to the visual domain, requiring models to evaluate the plausibility of hypothesis images based on a premise image and explain their decisions. NL-EYE consists of 350 carefully curated triplet examples (1,050 images) spanning diverse reasoning categories: physical, functional, logical, emotional, cultural, and social. Experiments show that VLMs struggle significantly on NL-EYE, often performing at random baseline levels, while humans excel in both plausibility prediction and explanation quality.

**Strengths:**

Previous Visual entailment tasks were mainly in text format. This paper for the first time proposes the task in image formats, and collected a human-curated benchmark. The experiments show that current VLMs cannot do well on the NL-EYE.
Also, one experiment result saying that VLM prediction depends on hypothesis location is interesting.

**Weaknesses:**

1. It is unclear whether the used prompt can best unleash VLMs' performance. For example, from Table 5, it seems no example has been provided, and that may lead to lower VLM performance.
2. Why do human only achieve 83-85% accuracy if human collected the dataset and this dataset do not require expert knowledge? (Line 426-427) It is a bit confusing to understand.
3. In Table 3, why not try GPT-4o as the Image-to-Text model? Also, why not try Claude models as predictor?
4. The images are generated instead of from real world, and could potentially affect the output. The test size is 350 which might be small.

**Questions:**

See Weakness 1-3.
Also just out of curiosity, why can't the setting in Table 3 solve this problem? E.g. How did GPT-4o fail the entailment upon the Figure 8 machine-generated captions?

---

> ### Author Response · Authors · 2024-11-18
>
> Thank you for your valuable feedback. We have thoroughly examined your comments and will begin by providing detailed clarifications and responses to each point.
>
> **Prompts**
>
> The prompt was manually optimized on a small subset of a few examples. Our choice to use a single, consistent prompt was to ensure a controlled evaluation environment. The goal was to isolate the performance differences between models rather than to find the best prompt for each model through prompt engineering.
>
> While analyzing different prompts can provide valuable insights, it is important to note that a well-performing VLM (or any model) should be able to understand and follow instructions. Thus, if a model is overly sensitive to input prompts, it suggests that it struggles with the task. To assess this, we are currently conducting experiments using 3 additional prompts. We currently report preliminary results and plan to include an ablation section in the paper.
>
> Results on GPT-4 Vision:
>
> | Prompt Variant   | Prompt Template Change        | Triplet Acc. (Separated) (%) |
> |-------------------|-------------------------------|------------------------------|
> | Original          | --                            | 46%                          |
> | First Explain     | First explain, then predict   | 43%                          |
> | CoT               | Let’s think step by step      | 49%                          |
> | Role              | You are a causality expert    | 44%                          |
>
> As evident from the preliminary results, the CoT approach leads to an improvement of 3%, while, overall, the prompts demonstrate comparable performance.
>
> **Human performance**
>
> The goal of the benchmark is to challenge the SOTA VLMs capabilities while keeping the task relatively intuitive for humans.  Similar to the human performance in NL-Eye, the human performance in recent test-set benchmarks such as Visual Riddles [7] and Winoground [1], is reported at 82% and 85.5%, respectively.
>
> In addition, we would like to note that the nature of the questions in our benchmark is not deterministic (i.e., it’s not a straightforward “Is it a cat or a dog?” type of question; as demonstrated in Figures 1, 2, and 3). Instead, people are asked to create a narrative that explains sequences of events and then assess which narrative is more plausible. This involves subjective interpretation, as individuals may perceive plausibility differently, leading to minor disagreements that reflect the complexity of abductive reasoning in visual scenarios. As a toy example for further intuition consider a riddle. The fact that person A creates a riddle does not guarantee that person B will solve it. Therefore, an accuracy of 85% represents a strong performance, particularly considering that it is based on the majority vote agreement among three annotators.
>
> **Image-to-text models**
>
> Our decision not to use GPT-4o as the Image-to-Text descriptor stems from the assumption that when an LLM serves as both descriptor and judge, it may prefer text generated from its own distribution, potentially biasing the evaluation. The motivation for the Image-to-Text experiment is to assess the model's ability to "communicate" relevant image content effectively rather than evaluate predictor performance alone. Therefore, we focused on using multiple models as descriptors and selected GPT-4o as the judge, given its high performance in the Text-only experiment.
> That said, following your suggestion, we conducted the experiment using Claude as an additional judge::
>
> | Describer       | Prediction Triplet (%) |             |
> |------------------|-------------------------|-------------|
> |                 | GPT-4o                 | Claude 3.5  |
> | Gemini-1.5-Pro  | 29%                    | 50%         |
> | GPT-4 vision    | 32%                    | 44%         |
> | LLaVA 1.6       | 29%                    | 36%         |
> | BLIP 2          | 40%                    | 42%         |
> | Instruct BLIP   | 35%                    | 36%         |
>
> Interestingly, Claude demonstrates better judgment over image descriptions; however, its performance remains comparable to the "dumb-pixel" baseline. Thanks to this suggestion, we will incorporate this observation into the paper.

---

> ### Author Response · Authors · 2024-11-18
>
> **Real Images**
>
> Thank you for raising this point. The advantages of synthesizing the benchmark include the flexibility to simulate a wide variety of everyday scenes while maintaining both consistency and quality. Extracting desired triplet scenes from real-world sources, such as videos, is challenging, less efficient, and sometimes impossible if we want consistency between the premise and the false hypothesis, which is not part of the video.
>
> However, we recognize that the style of images—whether generated or realistic—could potentially influence the model’s performance. In light of your feedback, we are conducting an ablation study on a subset of 20 triplets (60 images) by presenting them in both their original format and as real images. These images are either extracted from online sources or created and photographed by us. We select samples from our benchmark that are straightforward to produce—i.e., they do not require the same individual across images, and the scenes are common (more nuanced examples are challenging to find online). These examples are simpler, and we observe higher results with these examples.
>
> We found GPT-4 Vision achieves 58% accuracy on real natural images compared to 68% on generated ones, suggesting that the performance gap is not rooted in the type of images used. This analysis allows us to assess the impact of visual realism on model performance, and we will include it in the paper.
>
> **Test set size**
>
> Recent efforts in vision-and-language evaluations have increasingly emphasized "quality over quantity" when assessing foundation models. For instance, datasets like Winoground [1] (CVPR 2022), comprising only 400 examples, have profoundly influenced vision-language model advancements. Similarly, other widely adopted datasets, including WHOOPS! [2] (ICCV 2023), LlaVA-Bench [3] (NeurIPS 2023), Visit-Bench [4] (NeurIPS 2024), ConTextual [5] (ICML 2024), VibeEval [6], and Visual Riddles [7] (NeurIPS 2024), feature 90, 500, 576, 500, 269, and 400 examples, respectively, and are key to VLM evaluation. As you’ve noted, aligning with this trend, our dataset—though relatively small—is a carefully crafted challenge set specifically designed to test the capabilities of multimodal large models, not for training or fine-tuning. This distinction is fundamental to understanding its purpose and role within the benchmark. Furthermore, future work could explore automating dataset creation, developing a specialized training set using a model, and employing NL-Eye as a dedicated test set to further evaluate model performance.
>
> **Question**
>
> In reasoning tasks within VLMs, we consider two key components: recognition and reasoning. In our image-to-text task, we examine both by evaluating multiple descriptor models alongside a single predictor model. The generated captions from image-to-text tasks highlight differences in the ability to include relevant details that aid in determining plausibility. Specifically, in the captions from Figure 8, Claude effectively captured key details from the image (e.g., whether there is a match or not on a dating app), enabling GPT-4 to succeed. Consider a caption that omits or misinterprets these critical details—it becomes impossible to accurately assess which scenario is more likely to have occurred or is likely to occur.
>
>
>
> [1] Thrush, T., Jiang, R., Bartolo, M., Singh, A., Williams, A., Kiela, D., & Ross, C. (2022). Winoground: Probing vision and language models for visio-linguistic compositionality. In Proceedings of the IEEE/CVF Conference on Computer Vision and Pattern Recognition (pp. 5238-5248).
>
> [2] Bitton-Guetta, N., Bitton, Y., Hessel, J., Schmidt, L., Elovici, Y., Stanovsky, G., & Schwartz, R. (2023). Breaking common sense: Whoops! a vision-and-language benchmark of synthetic and compositional images. In Proceedings of the IEEE/CVF International Conference on Computer Vision (pp. 2616-2627).
>
> [3] Liu, H., Li, C., Wu, Q., & Lee, Y. J. (2024). Visual instruction tuning. Advances in neural information processing systems, 36.
>
> [4] Bitton, Y., Bansal, H., Hessel, J., Shao, R., Zhu, W., Awadalla, A., ... & Schmidt, L. (2023). Visit-bench: A dynamic benchmark for evaluating instruction-following vision-and-language models. Advances in Neural Information Processing Systems, 36, 26898-26922.
>
> ‏
> [5] Wadhawan, R., Bansal, H., Chang, K. W., & Peng, N. (2024). ConTextual: Evaluating Context-Sensitive Text-Rich Visual Reasoning in Large Multimodal Models. arXiv preprint arXiv:2401.13311.
>
> [6] Padlewski, P., Bain, M., Henderson, M., Zhu, Z., Relan, N., Pham, H., ... & Tay, Y. (2024). Vibe-Eval: A hard evaluation suite for measuring progress of multimodal language models. arXiv preprint arXiv:2405.02287.
> ‏
>
> [7] Bitton-Guetta, N., Slobodkin, A., Maimon, A., Habba, E., Rassin, R., Bitton, Y., ... & Elovici, Y. (2024). Visual Riddles: a Commonsense and World Knowledge Challenge for Large Vision and Language Models. arXiv preprint arXiv:2407.19474.‏ NeurIPS 2024

---

### Official Review · Reviewer_fRsv · 2024-11-04

**Soundness:** 3
**Presentation:** 3
**Contribution:** 3
**Rating:** 6
**Confidence:** 2

**Summary:**

This paper proposes a benchmark for measuring visual abductive reasoning capability and explains the process of constructing this benchmark. It demonstrates that current multimodal language models lack visual abductive reasoning capability and introduces a novel aspect of verifying image-to-image entailment that has not been previously addressed.

**Strengths:**

- The paper is well-written and easy to read.
- The process of data collection and verification is systematic and meticulous.
- It intriguingly points out the shortcomings of existing visual language models (VLMs) in visual abductive reasoning, with experimental results to substantiate this claim.
- The paper proposes various experimental setups by combining or separating images, changing the order of images, which helps ensure fair testing.
- The benchmark effectively reveals multiple shortcomings of different VLMs, not only evaluating abductive reasoning but also highlighting issues with image location sensitivity and poor visual interpretation.
- Unlike traditional natural language inference (NLI) benchmarks, this approach offers a comprehensive evaluation of multiple aspects.

**Weaknesses:**

- The evaluation criteria are unclear and not well-defined. The use of automatic evaluation for explanations seems inadequate, and manual evaluation, while more accurate, is too costly and varies depending on the person.
- The definition of visual abductive reasoning capability remains unclear; it appears to evaluate abilities including visual interpretation, interpretation of multiple images, and natural language inference, covering a broad range of concepts that are not distinctly defined.

**Questions:**

- For the evaluation with this benchmark, it would be beneficial to have better metrics. Are there methods to quantify image order sensitivity? Could metrics be developed to measure visual understanding and linguistic abstract reasoning capabilities using various forms of input (Text-only, Image-to-Text, Image and Text, etc.)?

---

> ### Author Response · Authors · 2024-11-18
>
> We would like to start by appreciating your review and the list of strengths you found in our paper. Your feedback is important to us, and we will try to address your weaknesses and improve our manuscript accordingly.
>
> **Abductive Reasoning Definition**
>
> Thank you for raising this point. Abductive reasoning is indeed a complex skill that involves multiple sub-capabilities. While we believe it is well-defined in our paper in lines 115-145, given the opportunity, we will clarify and explicitly outline the sub-capabilities you mentioned to enhance understanding. Notice that we conducted experiments to isolate and evaluate different capabilities in our study (as noted in lines 317-327, Reasoning Approaches & Input Strategies paragraph).
>
> As you noted, this skill includes a range of concepts, from basic abilities such as visual understanding, detection, and tracking to more advanced ones like plausibility assessment, common sense, interpretation, and decision-making. We incorporate clear definitions of each in our paper.
>
>
> **Evaluation Criteria**
>
> We dedicated over half a page to describing the four evaluation criteria in Subsection 4.2. Additionally, in Appendix A, we provide details about the prompt used for the automatic evaluation, and in lines 861-876, we present a mathematical formulation of the accuracy measures. We also include results for random and “dumb” baselines to help interpret the evaluation outcomes.
>
> We kindly ask you to specify which aspects or criteria you found unclear, so we can elaborate on them and make the necessary revisions to improve the manuscript.
>
>
> **Automatic Evaluation of Explanations**
>
> The manual evaluation conducted with crowd workers, which ensures a higher degree of accuracy, does not present significant cost challenges due to the efficient protocol described in Subsection 4.2 (lines 348-356). Therefore, we use human-generated explanations as gold references in our automatic evaluation approach. Notably, these gold references are an additional contribution and can be utilized by other researchers for the automatic evaluation of future models.
>
> We acknowledge that using GPT-4o for automatic evaluation with gold-reference explanations has limitations, as it may not capture all plausible explanations beyond the gold set. However, using LLMs to evaluate other LLMs (LLM-as-a-judge) is a widely adopted method. In our setup, the judge model is presented with human-selected gold references, enhancing the accuracy of its evaluation. Without these references, the model would need to assess the validity of explanations through abductive reasoning – precisely the capability we aim to evaluate. Thus, while automatic evaluation has constraints, it still provides reliable scores for comparing VLMs. The correlation between our automatic and manual evaluations is 0.5, which is considered high. For instance, in the comprehensive study of [1], the average correlation for LLM-as-a-judge models is below 0.5 (see Table 1).
>
> Having said that, we acknowledge this valuable point and consider extending our hybrid method by incorporating multiple automatic evaluation (auto-eval) approaches using various LLMs as judges. We aim to explore this direction in the hope of developing a more robust evaluation framework.
>
>
> **Question: Metrics**
>
> We would like to highlight that, as part of our benchmark, we plan to release not only the images but also the gold image descriptions (lines 218-227). These descriptions are used for text-only experiments, where we evaluate the linguistic abstract reasoning capabilities of the models (see Table 3 Gold describer).
>
>
> Additionally, our metrics account for the models' sensitivity to order. For instance, the consistency accuracy metric (defined in lines 331-337) evaluates this by asking the model to predict which hypothesis is more plausible while presenting the hypotheses in different orders. The model is considered correct only if it consistently identifies the gold plausible hypothesis in both orderings. We provide a detailed analysis of the models' sensitivity to order in lines 493-500 and in Table 6.
>
> We believe that, with the right experimental setup, our metrics can effectively isolate and quantify each model capability, as demonstrated in our paper. Specifically:
> _Image Setups (lines 284-316):_ We include both pairs and triplets of images.
> _Reasoning Approaches (lines 317-320):_ We evaluate both vision-based and text-based reasoning methods. _Input Strategies (lines 321-328):_ We explore a range of input formats, including multiple images, combined images (all-in-one), Image-to-Text, and Text-only inputs. These comprehensive setups ensure a thorough assessment of the models across different reasoning and input configurations.
>
>
> [1] https://arxiv.org/abs/2406.18403

---

> > ### Comment · Reviewer_fRsv · 2024-11-26
> >
> > Hello,
> >
> > Thank you for the clear explanation. It has greatly helped my understanding. I truly think this benchmark is excellent and highly useful.
> >
> > The limitation I wanted to point out regarding automatic evaluation aligns with what you mentioned—specifically, the reliance on comparing outputs to golden references. However, after looking at Appendix A.1, I noticed that you’ve attempted to address this by categorizing scores into three levels, which is a great way to enhance the evaluation process.
> >
> > I’ve also been reflecting on how I might use this dataset. If I were to evaluate my own model, I would likely follow the experimental setup outlined in Table 2. Then, to investigate potential weaknesses, I would analyze whether the issues stem from textual reasoning or visual interpretation, similar to the process described in Table 3.
> >
> > That said, one thing I found slightly challenging is that all the results are represented solely as accuracy scores. Since accuracy can be interpreted differently depending on the experimental setup, it requires an extra layer of thought to fully understand the results. This made me think it might be helpful if the results in Table 3 were expressed in terms of textual reasoning capability or visual interpretation capability, rather than just accuracy. I believe this could make the findings more interpretable and easier to relate to specific model strengths and weaknesses. However, the current format is still very practical. This was just an idea I had.
> >
> > Overall, I think this is an incredibly valuable benchmark. Thank you again for your thoughtful response and for providing these clarifications.

---

> > > ### Author Response · Authors · 2024-11-27
> > >
> > > Thank you for engaging in the discussion and sharing your valuable feedback. We now have a clearer understanding of your suggestion to develop specific metrics for textual and visual reasoning. As demonstrated in Table 3, the primary failure point of current VLMs lies in interpreting visual images, identifying the relevant elements within each image, and understanding the relationships between images necessary for solving the task.
> > >
> > > We will incorporate a discussion in the paper on designing metrics and experimental setups for future analysis. For instance, one possible direction could involve decomposing each triplet example into a list of objects and relationships that need to be identified in order to solve it, and then providing a metric based on what the model successfully identified. This approach could indeed improve our understanding of why VLMs fail.
> > >
> > > However, we believe that developing and implementing these metrics constitutes a broader research direction that extends beyond the scope of the current paper. Following your advice, we will ensure this idea is addressed as a potential future work in our revised submission.
> > >
> > > We would greatly appreciate any consideration to raise the score to improve the chances of our paper being accepted.

---

### Author Response · Authors · 2024-11-21

We thank the reviewers for their valuable feedback. We are pleased that the reviewers acknowledged the novelty of our dataset, particularly in integrating abductive reasoning across multiple image scenes and the quality of our systematic data collection (fRsv). We are also glad that reviewers found our approach clear and original (972W, GcX5, LDjN) and appreciated our comprehensive evaluation of VLMs (fepb, 972W).


In response to the reviewers constructive suggestions, we have incorporated considerable changes into the updated manuscript, (attached in the updated PDF):


- **Added open-source VLMs** to expand our comparative analysis (L427-L428; Table 12).
- **Explored new prompting strategies**, including reverse process, Chain-of-Thought, and role-specific prompts, to investigate their impact on reasoning performance (L281; Tables 6, 14).
- **Conducted ablation analysis on real images** (online and photographed) to assess the effect of visual realism on model performance (L1241-L1255, Fig.9).
- **Introduced comparisons** of image-to-text performance using Claude as a judge (L486-487; Table 13).
- **Clarified definitions** (L140-141; Appendix A.1)  and **expanded the discussion** (Appendix B) on future directions for improving abductive reasoning in VLMs.


We deeply appreciate this recognition and have provided detailed responses to each reviewer’s comments below.

---

### Author Response · Authors · 2024-11-25

Dear reviewers,

As the discussion period comes to an end, we kindly ask you to consider our responses and the additional experiments we conducted.
We would be happy to discuss our responses further and hope they address any misunderstandings or concerns you may have raised.

---

### Meta-Review · Area_Chair_RRyS · 2024-12-18

**Metareview:**

This paper introduces NL-EYE, a benchmark designed to test visual abductive reasoning by requiring models to determine which of two images (hypotheses) better fits a given premise image. Unlike typical visual NLI tasks that rely on textual premises, NL-EYE uses purely visual input and spans diverse reasoning categories. Reviewers praised the careful data collection, the clear problem formulation, and the exploration of model weaknesses, noting that current VLMs struggle with abductive reasoning and even show sensitivity to image order. Although some raised concerns about evaluation metrics, prompt selection, and the small dataset size, the novelty and thoroughness of NL-EYE stand out. The paper opens a new avenue for assessing complex visual reasoning capabilities beyond conventional benchmarks. Given the overall positive assessment of the benchmark's conceptual clarity and its potential to spur new research directions, I recommend acceptance.

**Additional Comments On Reviewer Discussion:**

During the discussion, the authors addressed concerns about metrics and provided clarifications on evaluation protocols. While some points, such as better metrics, more detailed prompts, and larger datasets, remain for future work, the reviewers generally agreed that NL-EYE is a meaningful step forward. Considering the benchmark's potential impact on advancing visual abductive reasoning research, I lean toward accept.

---

### Decision · Program_Chairs · 2025-01-22

Accept (Poster)